# Temporal matching as an accounting principle for green electricity claims

Hanna F. Scholta [1] ✉ & Maximilian J. Blaschke [1,2]

Labeling electricity as green typically relies on annual volumetric matching of certificates. Recent studies have shown that hourly matching can improve the environmental effectiveness of green electricity procurement. Responding to the European Union's push for more transparent and reliable green products, we assess how stricter temporal matching affects green electricity claims. Using data from the European certificate and electricity market, we analyze quarterly, monthly, weekly, daily, and hourly matching. We find substantial seasonal and intra-day mismatches between green electricity supply and demand that remain hidden under annual matching. As certificates already allow monthly granularity, we propose a two-phase transition: short-term adoption of quarterly or monthly matching to reduce seasonal accounting distortions, followed by hourly matching to address increasing day-night disparities. Fully integrating storage systems into certificate schemes is crucial for this transition. Our findings inform current policy debates, such as the revision of the Greenhouse Gas Protocol Scope 2 Guidance.

Worldwide, an increasing number of consumers is voluntarily opting for green products. To ensure the credibility of such products, recent policy frameworks[1–4] and legislation proposals[5,6] of the European Union (EU) advocate for more transparency and reliability. In the case of electricity, as of now, providers typically have to prove that a consumption-equivalent volume of renewable energy has been produced over the course of the respective year[7,8]. For this purpose, they cancel energy attribute certificates (EACs)–such as Renewable Energy Certificates in the United States (US) or Guarantees of Origin (GOs) in Europe–which can be traded independently of the associated electricity. This cancellation approach, commonly referred to as annual volumetric matching, allows green electricity claims to be based on renewable production from a time and location different than that of consumption. A closer alignment between supply and demand is currently neither mandated by legislation[7,8] nor by relevant standards[9].

In recent years, however, the discussion around facilitating closer alignment–often referred to as granular matching - has gained traction[10]. This momentum is further reinforced by ongoing revisions of key standards and market reassessments, such as the Greenhouse Gas (GHG) Protocol Scope 2 Guidance[11] and the 2025 review of the European GO system[12]. Among others, recent studies have found that increased temporal matching can improve the environmental effectiveness of green electricity claims on the system level[13,14]. Yet, despite the growing emphasis on transparency and reliability in recent EU policy frameworks and legislative proposals, comparatively little attention has been paid to green claims from an accounting perspective. We therefore ask: How would stricter temporal matching affect voluntary green electricity claims? What implications might different temporal matching requirements have?

It is important to note that the transparency and reliability behind such claims may be judged differently depending on the applied accounting principles. In this study, we adopt a perspective that links green claims to the availability of renewable electricity at the time of consumption, given that global decarbonization efforts aim to ultimately align total energy demand with renewable supply at all times. This temporal alignment of renewable demand and renewable generation is of high importance, especially due to the intermittency of renewable generation and the absence of free energy storage. However, we note that our perspective may not be mistaken for a universal definition. Possible alternative accounting principles that could be used in assessing green claims include, for example, adherence to recognized standards, long-term decarbonization outcomes, or

[1]TUM School of Management, Technische Universität München, Munich, Germany. [2]Center for Energy and Environmental Policy Research, Massachusetts Institute of Technology, Cambridge, MA, USA. ✉e-mail: hanna.scholta@tum.de

investment effects. With the temporal perspective used within this study, green claims should reflect a consistent temporal alignment between supply and demand. While granular matching may also encompass a locational dimension, we focus on the temporal dimension within this study. Accordingly, we illustrate and discuss the implications of increasingly strict temporal matching for green electricity claims within the largest standardized voluntary green electricity market–the European GO market[10].

So far, voluntary green electricity markets have largely been assessed through the lens of environmental impact: Early studies highlighted the advantages of integrated, multi-country certificate markets and long certificate validity periods, pointing to reduced price volatility and greater market flexibility as important conditions for investment in renewable capacity[15]. Since then, a substantial body of literature has taken a more critical view, questioning the effectiveness of voluntary green electricity markets in driving additional renewable capacity[16–22] or reducing emissions[13,14,23–26], and drawing attention to risks of double counting and consumer misconceptions[19,21,27,28]. More broadly, calls have emerged to revise renewable energy accounting for a more accurate reflection of environmental impact[25,29,30].

Similarly, research on temporal matching has so far also focused on environmental impact: Enhancing a capacity expansion planning model, Xu et al.[13] show that hourly matching of green electricity generation and demand by corporate consumers can lead to substantial system-level carbon dioxide ($CO_2$) reductions in the US when combined with regional and additionality constraints. Moreover, they find these reductions to be significantly greater than under annual volumetric matching. At the same time, they observe that imposing hourly matching increases overall system costs, with costs rising further as more corporate consumers adopt such procurement strategies[13]. Similar findings are reported by Riepin and Brown[14] for Europe. A recent meta-study, building amongst others on the findings of the two previously mentioned modeling papers, highlights that, in contrast to annual matching, hourly matching through Power Purchase Agreements with newly constructed local renewable generators leads to greater reductions in system-wide emissions[26].

Industry is already moving ahead with its own initiative, Energy-Tag pushing towards the development of guidelines for 24/7 clean electricity procurement based on granular energy certificates[10]. Major corporations such as Google have begun integrating hourly matching into their operations[31]. Yet, academic literature so far only offers limited empirical evidence on the temporal interplay between green electricity demand and supply[32]. While prior work on temporal matching has focused on system-level impacts, empirical evidence on how stricter temporal matching affects the transparency and reliability of green electricity claims is scarce. We address these gaps by analyzing real-world GO data from 2016 to 2021 alongside European electricity market data on electricity demand and renewable supply to evaluate green electricity claims under increasingly strict temporal matching requirements.

In this work, we show that, from a temporal transparency perspective, annual matching conceals substantial seasonal and intra-day mismatches between renewable electricity supply and demand in the European GO market. We find that quarterly or monthly matching can help reduce seasonal distortions, while hourly matching may help to address growing day-night disparities. Furthermore, we suggest integrating storage systems into certificate schemes to achieve a higher temporal alignment within green claims. Our paper introduces an accounting perspective to the evolving discussion on granular matching and gives policy recommendations for transparent and reliable claims, particularly in the context of the current revision of the GHG Protocol Scope 2 Guidance[11]. In addition, our insights into the quarterly, monthly, weekly, daily, and hourly balances of supply and demand in the GO market not only contribute to the previously mentioned call for empirical studies on the temporal interplay between green electricity demand and supply[32], but may also prove valuable for the 2025 review of the European GO system[12].

## Results

### GO market characteristics and central assumptions

We begin the Results section with an overview of the GO market and the central assumptions that guide our approach, which is essential for interpreting the subsequent findings. The European GO market was initially introduced in 2001[33]. In 2009, GOs were legally defined as "an electronic document [...] providing proof to a final customer that a given share or quantity of energy was produced from renewable sources"[34]. Renewable energy producers may receive such a document per megawatt hour (MWh) of renewable electricity generation, and its reception is commonly referred to as GO issuance. Energy released from storage systems may only issue GOs if the respective systems are located next to a renewable energy generation facility and no GOs have been issued by that facility[35]. Within the same year, the EU stipulated that the disclosure of the share of electricity from renewable sources over the preceding year should be done using GOs[36]. The act of using a GO for a green claim is referred to as GO cancellation. Any GOs that have not been canceled 18 months after issuance expire[7]. GO certificates include a timestamp indicating the time of electricity production, which typically refers to the month of generation[7]. While the 2023 amendment of the EU's Renewable Energy Directive (RED III)[12] introduced the possibility of issuing GOs with (sub-)hourly timestamps, most GOs currently continue to rely on monthly timestamps. Providing an electronic hub, the Association of Issuing Bodies (AIB) sits at the interface of GO-trade within Europe. By 2021, its hub connected 24 European countries[37], which in total issued GOs for 748 TWh of green electricity. In the past, the market has shown a structural oversupply[20–22]. Unlike in mandatory green electricity markets, where oversupply can be regulated through government-imposed quota adjustments[15], no such mechanism exists in voluntary markets as demand depends on voluntary commitments by market participants (see Supplementary Note for further background on voluntary and mandatory markets). Over the last years, however, annual GO supply and demand have become increasingly balanced (from 26% over-supply in 2016 to merely 3% in 2021)[38,39]. Although the focus is frequently put on corporate consumers taking advantage of reporting zero emissions for their electricity consumption[13,14,23,24,29], demand for green electricity has increased across sectors[19–22,40]. In the German residential sector, for instance, the share of green electricity purchased has risen from 14% to 43% over the past decade[41].

In the context of this paper, annual green electricity supply refers to the GOs issued within a year, while annual green electricity demand refers to the canceled GOs issued in the respective year. To contrast green electricity supply and demand on an hourly level, we interpolate hourly GO-demand and hourly GO-supply data by assuming a direct correlation with overall European electricity consumption and renewable generation. For example, we use data on the hourly renewable electricity generation to draw conclusions on the hourly supply of GOs. However, not all renewable generation also issues GOs (about 60% in 2020)[42]. Member states can decide not to issue GOs for renewable energy installations that receive financial support[7]. Since different renewable energy sources depend on financial support to varying degrees[21], there are differences in the composition of energy sources between GO-issuing generation and total renewable electricity generation. Most noteworthy is that the share of hydropower within the GO-issuing generation is disproportionally large. As a result, the generation pattern of total renewable electricity generation partially differs from that of GO-issuing generation. However, both generation profiles show the same trend: a decline in the dominance of hydro-power in favor of wind and solar, more non-dispatchable, variable renewable energy[37–39,43,44]. Using the pattern of total renewable electricity generation, hence, allows us to study the implications of stricter

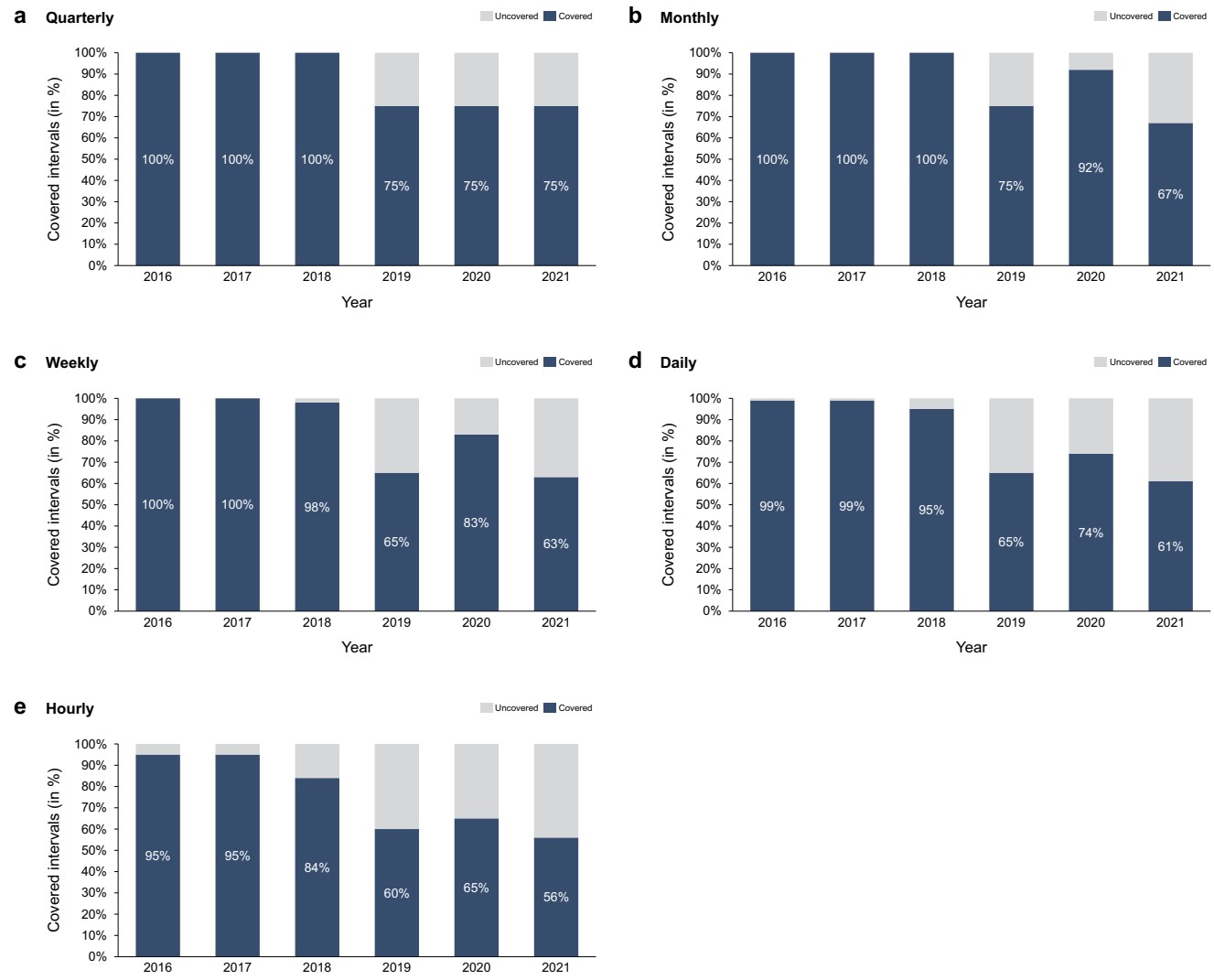

**Fig. 1 | Demand intervals covered and uncovered with green electricity supply under increasingly strict temporal matching.** This figure shows the share of intervals in which green electricity demand was (un-)covered by green electricity supply between 2016 and 2021 under increasingly strict temporal matching. Panels display results for matching on a quarterly (**a**), monthly (**b**), weekly (**c**), daily (**d**) and hourly (**e**) basis. Dark blue bars indicate the proportion of intervals where demand was fully covered, while light grey bars represent intervals with uncovered demand. The imposition of stricter temporal matching reveals shortfalls of supply relative to demand, and these shortfalls increase over the years. In addition, the share of covered intervals decreases as requirements become stricter.

temporal matching on green electricity claims against the backdrop of the shifting energy mix for future energy scenarios.

On the demand side–aiming to reflect all sectors' involvement in the sourcing of green electricity[19–22,24,40]–we approximate the hourly demand of GOs according to total load data. Thereby, we assume all sectors to consume green electricity in a constant manner and proportion of their electricity consumption (see Methods for details on our methodological approach).

**Uncovered intervals under stricter temporal matching**

In line with current regulations, green electricity supply consistently meets green electricity demand when accounting on annual volumetric basis. Introducing more granular matching reveals significant discrepancies (see Fig. 1). Unsurprisingly, the stricter the imposed temporal matching principle, the more frequently the supply falls short of the demand. However, the share of intervals with insufficient coverage has increased over the years, reflecting the growing demand for green electricity that also reduces oversupply on the annual level, as well as the shift in the energy mix toward more and more variable

renewable energy. The only year that falls slightly out of line is 2020, likely due to the impact of COVID-19 that made 2020 the only year in which the overall demand growth fell short behind the supply growth. Moving forward, we expect the green electricity demand to grow further as electrification for decarbonization measures (e.g., heat pumps or electric vehicles) continue to advance, marked by an increased coupling of the electricity with the transport, building, and industrial sector[45,46].

Under quarterly matching, one out of four quarters–either Quarter 1 or Quarter 4–would have shown insufficient coverage in recent years. On average, 1.2% of the demand remained uncovered in Quarter 1, while Quarter 4 experienced a 2.6% shortfall. Monthly matching sheds further light on the months at risk within the quarters, with January, February, September, October, and November being particularly prone to shortages. As evident in 2020, even a single month with a significant deficit can be sufficient to cause a quarterly shortage if the other months of the respective quarter do not compensate with an equivalent over-coverage. While October registers the lowest average shortage (2.4%), with 6.5%, November shows the

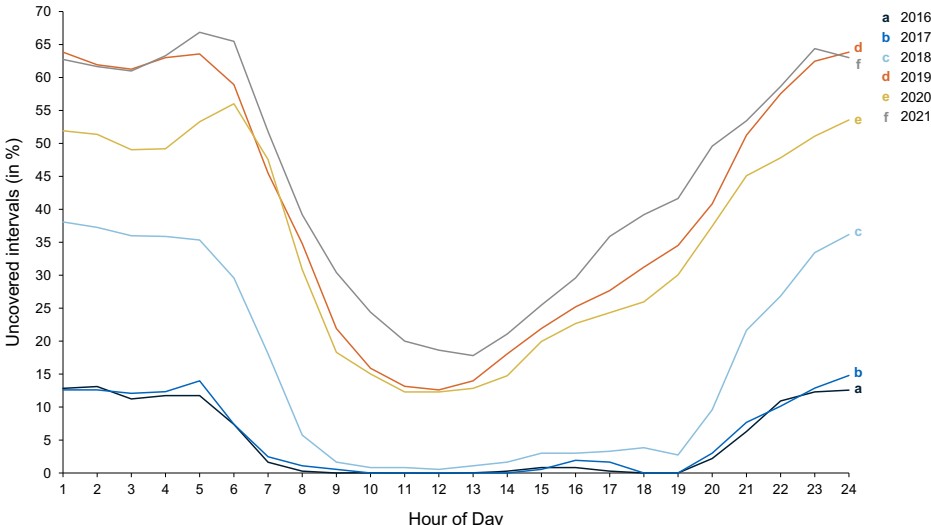

**Fig. 2 | Distribution of hours with a shortage in green electricity supply over the day.** This figure depicts the share of hourly intervals with unmet demand across the day for each year from 2016 to 2021. In 2016 (line a) and 2017 (line b), shortages were largely absent during daytime, whereas by 2021 (line f), they can also be observed around midday. Shortages are generally more frequent at night, with the disparity between day and night widening over time. The 2021 curve (line f) illustrates the strongest day-night disparity.

highest. December appears less likely to experience shortages due to comparatively low demand—potentially due to large manufacturing companies pausing operations during the Christmas holiday break (see Supplementary Fig. 1 for a more detailed overview of uncovered quarters and months per year).

By counting uncovered intervals across increasingly strict temporal matching requirements, our analysis reveals how claims under annual volumetric matching fail to account for seasonal fluctuations in renewable energy generation. This exposes the shortcomings of such claims from a temporal matching viewpoint. As illustrated in Fig. 1, the strongest mismatch between supply and demand occurs under hourly matching.

A closer examination of the hourly matching scheme reveals two major trends (see Fig. 2): Firstly, the share of uncovered intervals has increased over the years. In 2016, shortages from 7 a.m. to 7 p.m. have been almost non-existent. By contrast, in 2021, even during peak solar production hours like 1 p.m., shortages are observed in 18% of the intervals. Secondly, the night hours exhibit more shortages than the day hours. This disparity between day and night hours has grown more pronounced over the years, with the range of uncovered intervals expanding significantly—from a 13 to a 49 percentage-point difference. For instance, in 2021, the share of uncovered intervals peaks at 66.8% at the fifth hour of the day. Notably, this hour of the day, on average, also records the largest volumetric shortfall in meeting green electricity demand with supply (11.4%).

While the overall increase in the share of intervals with insufficient coverage can be attributed to the development toward a more balanced market on the annual level, the increasing day-night disparities underline how the growing share of variable renewable energy, particularly solar and wind, is outweighing the smoothening effects of dispatchable renewable energy sources like hydropower (see Supplementary Fig. 2 for further analysis).

We expect the day-night disparity to become even more pronounced in the future since natural limits prevent a further expansion of more hydropower in Europe[19]. The EU's plan to become climate neutral by 2050[47], thus, builds heavily on a further expansion of non-dispatchable renewable energy. In comparison to 2020, at least a doubling in onshore wind capacity and a quadrupling in solar and in offshore wind capacity is planned[48–51]. Today, the GO-issuing electricity generation may still show a higher representation of hydropower than

the actual renewable generation in Europe, which we used to approximate the hourly supply. Consequently, the day-night disparities in the European GO market may, at the moment, be weaker than what our results would imply. In the long run, however, we expect the increased reliance on solar and wind to lead to day-night disparities as we observe them. In addition, decarbonization efforts in the transportation sector may further accelerate the day-night disparity as electric vehicles are likely to increase electricity demand particularly in the evening hours and at night due to user charging preferences[52].

## Implications of temporal matching for green claims

An analysis of the changes in the volumetric scale of under-coverage under the imposition of stricter temporal matching further confirms the two main trends previously observed through the metric of uncovered intervals. First, the overall scale of under-coverage has also increased over the years. In 2016, only 0.2% of the annual green electricity demand remain uncovered under hourly matching. In 2021, this increased to 0.7% under quarterly matching and 4.3% under hourly matching. Additionally, the day-night disparity in volumetric under-coverage has also intensified (see Supplementary Fig. 3 and Supplementary Table 1).

To further assess the volumetric under-coverage, we estimate the increase in solar and wind generation required to close the supply-demand gap under different temporal matching requirements, assuming inelastic demand. We focus on wind and solar as the EU's expansion plans are centered on the large-scale deployment of these technologies. However, because the GO market is voluntary, stricter requirements—and potentially increasing cost of making green electricity claims—could exceed some consumers' willingness to pay, potentially leading to market dropouts and a reduction in overall demand. Therefore, the estimated increase in solar and wind generation should be interpreted with caution and not mistaken for a projection of future capacity additions resulting from stricter temporal matching requirements. Rather, these estimates serve to better understand the magnitude of the mismatches between green electricity supply and demand.

Figure 3 illustrates the scale-up in solar and wind generation required to cover the most undersupplied intervals under different temporal matching requirements in 2021. While the volumetric under-coverages may appear small relative to total annual green electricity

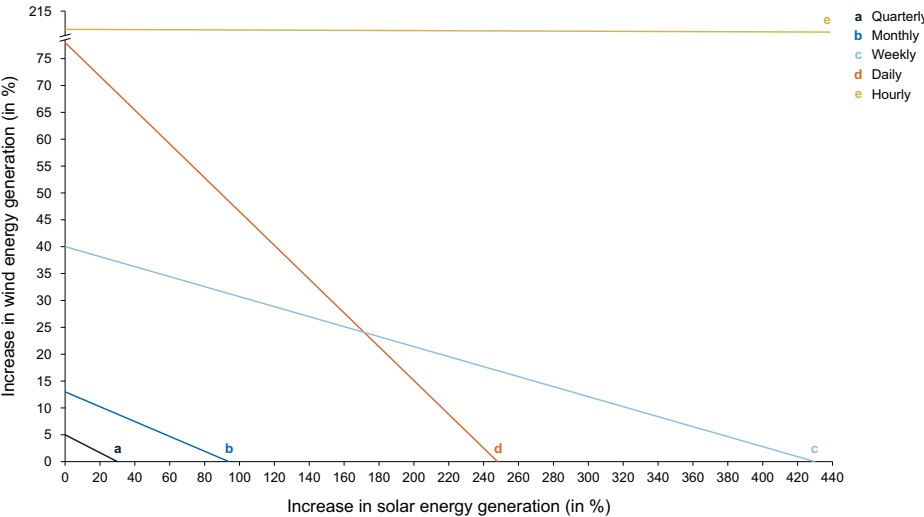

**Fig. 3 | Required scale-up in solar and wind generation to cover the most undersupplied intervals in 2021 under different matching schemes.** This figure illustrates the additional solar and wind generation needed to cover the most undersupplied intervals in 2021, expressed as a share of 2021 solar and wind generation. Each line represents one temporal matching scheme: quarterly (line a), monthly (line b), weekly (line c), daily (line d) and hourly (line e). Stricter requirements shift the curves outward. The yellow line for hourly matching illustrates how ineffective increases in solar generation are for overcoming the day-night disparity. Note: the large-scale solar increase suggested by the nearly horizontal nature of the line is due to minimal solar production at night in our dataset, resulting from the large geographic region covered in our analysis and differences in sunrise and sunset times. As solar panels inherently require sunlight to generate electricity, it is not part of a feasible solution to overcome night shortfalls.

demand, this analysis shows that overcoming these mismatches—if approached through additional solar and wind generation—would require a substantial increase in generation due to the inherent variability of these sources.

Assuming inelastic green electricity demand, addressing the most undersupplied intervals under quarterly matching would already require either a 5% increase in wind generation, a 30% increase in solar generation, or a combination of both sources as indicated with the straight lines between both axes. Under hourly matching, a substantial increase in wind generation (211%) would be required. An increase in solar generation, regardless of its scale, is no longer viable as it cannot resolve nighttime shortages due to its reliance on daylight.

## Discussion

As shown in "Results", stricter temporal matching reveals substantial seasonal and intra-day mismatches between green electricity supply and demand that remain concealed within green electricity claims under annual volumetric accounting. A stable sustainable energy system, however, ultimately depends on balancing renewable supply and demand continuously throughout the year, rather than only on an annual aggregate basis. Our results indicate that this principle is not adequately reflected in current accounting practices, and we therefore find current green electricity claims to be insufficiently transparent and reliable from a temporal matching viewpoint. Moreover, since GOs already carry monthly timestamps[7], matching schemes that account for seasonal fluctuations could likely be implemented without substantial modifications to existing systems. France, for instance, already uses monthly matching[53], and Switzerland plans to adopt quarterly matching by 2027/2028[54], demonstrating the feasibility of more granular approaches within existing frameworks. From a temporal transparency perspective, we find limited justification for adhering to annual matching as transitioning to quarterly or monthly matching could help overcome seasonal accounting distortions. However, our findings also indicated a growing day-night disparity that would remain hidden under quarterly or monthly matching. Moving toward hourly granularity in tracking and matching practices, thus, appears to become increasingly important for achieving transparent and reliable claims from a temporal matching viewpoint. While the EU has taken a first step in this direction by introducing the option of issuing (sub-) hourly timestamped GOs with the 2023 amendment of RED III[12], most European certificates still carry monthly production timestamps. We therefore recommend a phased approach for transitioning to stricter temporal matching: in the long term, we perceive hourly matching as essential to achieve temporally transparent and reliable claims. However, in the short term, quarterly or monthly matching could already help enhance green electricity claims while leveraging the capabilities of existing systems.

Our findings in relation to hourly matching are consistent with those of Xu et al.[13] and Riepin and Brown[14], who point to the system-level benefits of hourly matching, including greater $CO_2$ reductions and more diversified clean energy deployment. They also align with the argument of de Chalendar and Benson[25], who call for carbon accounting approaches that differentiate between renewable technologies depending on the grid mix at specific times of day, noting, for example, that wind capacity can deliver greater carbon reduction benefits than solar in solar-dominated grids. Similarly, the meta-study by Langer et al.[26] finds that hourly matching can deliver greater emission reductions than annual matching. While our analysis focuses on the transparency and reliability of green electricity claims rather than system-level impacts of green electricity procurement, it similarly underscores the importance of aligning how claims are accounted for with the temporal availability of renewable generation. The derived recommendation of a phased approach for introducing stricter temporal matching also appears reasonable against the backdrop of the significant cost premium that can be associated with hourly matching[13,14], as well as anticipated advances in digitization and data availability that could further ease the transition[55].

The findings presented in this paper further suggest that stricter temporal matching could affect price dynamics within voluntary green electricity markets, provided that participants continue to engage despite the increased complexity and potential cost implications. Stricter requirements could increase certificate scarcity during winter quarters, thereby raising their relative value and strengthening price signals that favor technologies less affected by seasonal variation, such as wind energy. Under hourly matching, time-stamped certificates may introduce sharper temporal price differentiation, increasing the

relative value of EACs issued during night or early morning hours when renewable generation is typically low. On the demand side, such signals could incentivize consumers to shift flexible loads[56] toward hours with lower EAC prices and higher renewable availability. On the supply side, they may provide attractive revenue opportunities for generation strategies designed to mitigate shortages, such as west- or east-facing PV units, while also improving the economics of energy storage. So far, storage is generally most profitable in systems with high shares of variable renewables[57,58]. Fully integrating it into the EAC value chain could open additional arbitrage opportunities. Therefore, storage systems should be allowed to issue EACs for discharged electricity, provided that a corresponding volume of EACs from electricity generated at the time of charging has been canceled. Industry initiatives are moving in this direction, with EnergyTag having recently introduced a framework for the issuance and cancellation of granular certificates related to storage[59].

Future research could assess whether, and to what extent, temporal matching translates into measurable system-level outcomes in voluntary settings. Among other things, this requires further examination of demand elasticity and willingness to pay, as these are critical for mitigating the risk of market disengagement. In this context, it may also be valuable to study the interplay between stricter temporal and locational matching. While temporal matching preserves market flexibility by allowing cross-border certificate trading, this could be further reduced if combined with locational matching. Moreover, mechanisms to incentivize adherence to more granular matching requirements in voluntary settings require further exploration. Additional work could further provide cost estimations on the different levels of temporal alignment, along with in-depth analyzes of the feasibility of more granular matching for different participants. While our analysis centers on voluntary green electricity markets, future research could examine how stricter temporal matching might contribute to the evolution of mandatory market frameworks.

## Methods
### GO data and European electricity market data merge
For our analysis of green electricity claims under increasingly strict temporal matching requirements, we draw on historical GO and electricity market data from 24 European GO trading countries from 2016 to 2021. Therefore, we combine data from the Association of Issuing Bodies (AIB) and the European Network of Transmission System Operators for Electricity (entso-e) to calculate the hypothetical quarterly, monthly, weekly, daily, and hourly coverages of green electricity demand by green electricity supply. The AIB reports statistics on historical GO issuance, cancellation, and trade of GOs. Entso-e provides up to quarter-hourly data on historical electricity generation and consumption within Europe[60,61].

From the AIB production statistics[38,39] (as the AIB changed the format of the statistics in 2019), we used the statistics in the old format for data on the years 2016–2018 (Tab "Monthly - Fuel"), and the statistics in the new format from 2019 onwards (Tab "All statistics"), we derive data on the cumulative monthly GO issuance per production type from 2016 to 2021 aggregated over all AIB member states connected via the AIB hub (GO-supply). We also use the AIB production statistics for yearly cumulative data over the respective AIB member states on the GOs issued and canceled within each year from 2016 to 2021 (GO-demand). Considering only GOs issued for renewable production, we omit all other types of GOs. For an overview of the countries included in our analysis, please see Supplementary Table 2.

To increase the granularity of the AIB data on GO-supply, we use entso-e data on the actual generation per production type for the years 2016–2021[62]. Depending on the country, the generation data points refer to different measurement intervals (15 min, 30 min, or 60 min). As our analysis requires an hourly frequency, we harmonize the different measurement intervals. By using the mean of all measurements within a respective hour, we derive a data set of the hourly electricity generation per production type and country. Subsequently, we aggregate the generation per country over all renewable sources that could have led to the issuance of renewable GOs. For the few missing values within the data set, we apply a structured data imputation strategy (see section Missing value imputation in European electricity market data). We then aggregate the data on renewable generation per hour for all countries that, at the respective time, had been connected via the AIB hub and were, thus, reflected in the AIB statistics. Since GOs have not been issued for every MWh of renewable electricity generation, the monthly GO-supply only makes up a fraction ($a_{m,y}$) of the monthly renewable electricity generation ($reg_{m,y}$). Hence, to finally derive the hourly green electricity supply (ge-supply), we scale each hourly value in the modified entso-e data set with $a_{m,y}$ (see Eq. (1)).

$$\text{ge} - \text{supply}_{h,d,m,y} := \text{GO} - \text{supply}_{h,d,m,y} = a_{m,y} \cdot \text{reg}_{h,d,m,y} \quad (1)$$

where,

$$h \in H := \{1, 2, ..., 24\}$$
$$d \in D_{m,y} := \text{All days of month } m \text{ in year } y$$
$$m \in M := \{1, 2, ..., 12\}$$
$$y \in Y := \{2016, 2017, ..., 2021\}$$

Similarly, we use entso-e data on the actual total load for the years 2016–2021[63] to derive the hourly multi-sectoral green electricity demand (ge-demand). Due to the differences in resolution, we also harmonize the data to an hourly level per country. For the missing values, we apply a structured data imputation strategy (see section Missing value imputation in European electricity market data). After artificially imputing the missing data, we aggregate the consumption data per hour for all countries that were connected via the AIB hub at the respective time. Since doing so, we consider the overall consumption of electricity (ec), instead of solely that of green electricity, also here the annual GO demand only makes up a fraction ($b_y$) of the yearly electricity consumption ($ec_y$). Scaling each hourly demand value in the modified entso-e data set by means of $b_y$, we derive the hourly ge-demand as follows:

$$\text{ge} - \text{demand}_{h,d,m,y} := \text{GO} - \text{demand}_{h,d,m,y} = b_y \cdot \text{ec}_{h,d,m,y} \quad (2)$$

where,

$$h \in H := \{1, 2, ..., 24\}$$
$$d \in D_{m,y} := \text{All days of month } m \text{ in year } y$$
$$m \in M := \{1, 2, ..., 12\}$$
$$y \in Y := \{2016, 2017, ..., 2021\}$$

### Missing value imputation in European electricity market data
While the entso-e data is very granular and comprehensive data on European electricity generation and consumption, it also shows some limitations—not only does it lack Icelandic data for both the generation and the demand side, but it also shows sporadic data gaps in the other 23 countries (two major gaps larger than a month and multiple minor gaps). The lack of Icelandic data originates from the lack of any physical connections to other entso-e members. As, however, Iceland was connected to the AIB-hub for the years of our analysis, engaging in the issuance and cancellation of GOs, we require its electricity generation and consumption data despite the absence of any physical connections. We impute the data for Iceland and the other sporadic data gaps (in total 7.2% of the data points contributing 2.4% of the renewable electricity generation and 5.5% of the data points contributing 0.9% of

the electricity consumption) by applying a structured and transparent data imputation strategy. We, therefore, draw on related values within the data set and incorporate additional data from other providers such as the International Energy Agency (IEA)[64] and the Statistical Office of the European Union (EUROSTAT)[65]:

Due to the lack of any other provider of hourly data for the two major gaps (renewable generation in Croatia (HR) from 2016 to 2018 and electricity consumption in Cyprus (CY) from January to September 2016), we impute the missing data by using scaled generation and consumption data from the time after the data gap. For Croatia, we use the net electricity production data from the IEA's Monthly Electricity Statistics[64], derive the required scaling factors by putting the monthly Croatian 2019 total renewable generation in relation to the 2016 to 2018 total renewable generation, and, finally, calculate the missing hourly values displayed in Eq. (3).

$$\mathrm{reg}(\mathrm{HR})_{h,d,m,y} = \frac{\mathrm{reg}_{\mathrm{IEA}}(\mathrm{HR})_{m,y}}{\mathrm{reg}_{\mathrm{IEA}}(\mathrm{HR})_{m,2019}} \cdot \mathrm{reg}_{\mathrm{entso-e}}(\mathrm{HR})_{h,d,m,2019} \quad (3)$$

where,

$$h \in H := \{1, 2, \dots, 24\}$$
$$d \in D_{m,y} := \text{All days of month } m \text{ in year } y$$
$$m \in M := \{1, 2, \dots, 12\}$$
$$y \in Y := \{2016, 2017, 2018\}$$

As the IEA's Monthly Electricity Statistics do not feature Cypriot electricity consumption data, we use 2016 and 2017 data from EUROSTAT on the electricity available to the internal market for Cyprus[65]. We derive the scaling factors based on the relation of the Cypriot monthly electricity consumption in 2016 to that in 2017 and calculate the hourly values as follows:

$$\mathrm{ec}(\mathrm{CY})_{h,d,m,2016} = \frac{\mathrm{ec}_{\mathrm{EUROSTAT}}(\mathrm{CY})_{m,2016}}{\mathrm{ec}_{\mathrm{EUROSTAT}}(\mathrm{CY})_{m,2017}} \cdot \mathrm{ec}_{\mathrm{entso-e}}(\mathrm{CY})_{h,d,m,2017} \quad (4)$$

where,

$$h \in H := \{1, 2, \dots, 24\}$$
$$d \in D_{m,y} := \text{All days of month } m \text{ in year } y$$
$$m \in M := \{1, 2, \dots, 9\}$$

For the minor gaps within the data set, we apply a two-pronged approach: Wherever solely one timestamp is missing, we linearly interpolate its value with the mean of the value of the data point before and after the gap (approach A). If more than one data point is missing, we use the mean of the values of the closest available surrounding data points at the same time of day (approach B).

For the missing Icelandic data, we use the generation and consumption of selected reference countries as a proxy, scaling them to the Icelandic level: Iceland primarily builds on hydro and geothermal sources[64]. We, therefore, use Norwegian hydro and Italian geothermal generation data as a reference, since these countries show an equally high share of hydro/geothermal generation in their renewable electricity generation[64] (we use the aggregated Norwegian hydro generation ("Hydro Pumped Storage", "Hydro Run-of-river and poundage", "Hydro Water Reservoir" and "Marine") and the Italian geothermal generation ("Geothermal") per hour). Besides, we also use the data on Norwegian electricity consumption. We address gaps in the data of both generation subsets by applying the same two-pronged approach also used for the total renewable generation and electricity consumption data per country. Drawing on the Monthly Electricity Statistics of the IEA[64], we derive time-dependent scaling factors by putting the IEA data of the reference countries in contrast to the IEA Icelandic data (using the following IEA data: "Net electricity production Hydro"

**Table 1 | Impact of data imputation in relation to total renewable electricity generation and electricity consumption (in %)**

|  |  | Generation | Consumption |
|---|---|---|---|
| Iceland |  | 1.94 | 0.81 |
| Major | Croatia | 0.43 | / |
|  | Cyprus | / | 0.02 |
| Minor | Approach A | 0.01 | 0.02 |
|  | Approach B | 0.03 | 0.05 |
| Total |  | 2.41 | 0.91 |

in Iceland and Norway for scaling the modified entso-e Norwegian Hydro generation data, "Net electricity production Geothermal" in Iceland and Italy for the modified entso-e Italian geothermal generation data, and "Final Consumption Electricity" in Iceland and Norway for the modified entso-e Norwegian electricity consumption data). We finally calculate the hourly Icelandic values for renewable electricity generation and electricity consumption as follows:

$$\mathrm{reg}(\mathrm{IS})_{h,d,m,y} = \left(\frac{\mathrm{reg}_{\mathrm{IEA}}(\mathrm{IS,hydro})_{m,y}}{\mathrm{reg}_{\mathrm{IEA}}(\mathrm{NO,hydro})_{m,y}}\right) \cdot \mathrm{reg}_{\mathrm{entso-e}}(\mathrm{NO,hydro})_{h,d,m,y}$$
$$+ \left(\frac{\mathrm{reg}_{\mathrm{IEA}}(\mathrm{IS,geothermal})_{m,y}}{\mathrm{reg}_{\mathrm{IEA}}(\mathrm{IT,geothermal})_{m,y}}\right) \cdot \mathrm{reg}_{\mathrm{entso-e}}(\mathrm{IT,geothermal})_{h,d,m,y} \quad (5)$$

$$\mathrm{ec}(\mathrm{IS})_{h,d,m,y} = \frac{\mathrm{ec}_{\mathrm{IEA}}(\mathrm{IS})_{m,y}}{\mathrm{ec}_{\mathrm{IEA}}(\mathrm{NO})_{m,y}} \cdot \mathrm{ec}_{\mathrm{entso-e}}(\mathrm{NO})_{h,d,m,y} \quad (6)$$

where,

$$h \in H := \{1, 2, \dots, 24\}$$
$$d \in D_{m,y} := \text{All days of month } m \text{ in year } y$$
$$m \in M := \{1, 2, \dots, 12\}$$
$$y \in Y := \{2016, 2017, \dots, 2021\}$$

Table 1 provides a conclusive overview of which share of total renewable electricity generation and electricity consumption within the final data set is based on which type of data imputation.

## Coverage calculation under stricter temporal matching

We finally calculate the green electricity coverages by contrasting ge-demand with ge-supply at an hourly level (see Eq. (7)). Subsequently, we aggregate the coverages to a daily, weekly, monthly, and quarterly level.

$$\mathrm{ge-coverage}_{h,d,m,y} := \mathrm{ge-supply}_{h,d,m,y} - \mathrm{ge-demand}_{h,d,m,y} \quad (7)$$

where,

$$h \in H := \{1, 2, \dots, 24\}$$
$$d \in D_{m,y} := \text{All days of month } m \text{ in year } y$$
$$m \in M := \{1, 2, \dots, 12\}$$
$$y \in Y := \{2016, 2017, \dots, 2021\}$$

## Further analyses on the role of solar and wind power

Given the growing significance of variable renewable energy sources in green electricity generation, we conduct further analyzes on the roles of solar and wind power. To examine their role in the trends observed at the aggregate level, we additionally calculate the hourly coverage of ge-demand by ge-supply across three hypothetical generation scenarios: one relying solely on solar, one solely on wind, and one combining both production types. We further assess the potential scale-up in solar or wind

generation needed to cover the most under-supplied intervals under each temporal matching requirement to better understand the implications of the mismatches.

For solar-only generation, we filter the entso-e data for the production type "solar" and harmonize the different measurement intervals to an hourly level. We complement missing values with our structured data imputation strategy outlined in section Missing value imputation in European electricity market data. For reasons of simplicity, we ease the threshold value between minor and major data gaps from 1 to 2 months and use the same scaling factors as before for the Croatian data gap in this analysis as imputed values only make up 0.05% of total solar generation. We aggregate the solar generation per hour for all countries that, at the respective time, were connected via the AIB hub. Considering the relation between total solar generation and GO-issuing solar generation ($a(\text{solar})_{m,y}$) and total electricity consumption and solar GO cancellations ($b(\text{solar})_y$), we calculate the hourly supply and demand values. For wind-only generation, we proceed the same way but initially filter the entso-e data for the production types "wind offshore" and "wind onshore" instead. Imputed MWhs here also only make up a small fraction (0.4%).

We calculate the coverages for all intervals at an hourly level as displayed in Eqs. (8)–(10) for solar exemplarily.

$$\text{ge−coverage}(\text{solar})_{h,d,m,y} := \text{ge−supply}(\text{solar})_{h,d,m,y} - \text{ge−demand}(\text{solar})_{h,d,m,y} \tag{8}$$

$$\text{ge−supply}(\text{solar})_{h,d,m,y} := \text{GO−supply}(\text{solar})_{h,d,m,y} = a(\text{solar})_y \cdot \text{reg}(\text{solar})_{h,d,m,y} \tag{9}$$

$$\text{ge−demand}(\text{solar})_{h,d,m,y} := \text{GO−demand}(\text{solar})_{h,d,m,y} = b(\text{solar})_y \cdot \text{ec}_{h,d,m,y} \tag{10}$$

where,

$$h \in H := \{1, 2, ..., 24\}$$
$$d \in D_{m,y} := \text{All days of month } m \text{ in year } y$$
$$m \in M := \{1, 2, ..., 12\}$$
$$y \in Y := \{2016, 2017, ..., 2021\}$$

Besides, we calculate the hourly coverages for the hybrid scenario as follows:

$$\text{ge−coverage}(\text{solar \& wind})_{h,d,m,y} = \left(\text{ge−supply}(\text{solar})_{h,d,m,y} + \text{ge−supply}(\text{wind})_{h,d,m,y}\right) - \left(\text{ge−demand}(\text{solar})_{h,d,m,y} + \text{ge−demand}(\text{wind})_{h,d,m,y}\right) \tag{11}$$

To better understand the potential implications of supply-demand mismatches, we use the ratio of the coverage level of the most undersupplied interval (min_ge-coverage) under each temporal matching requirement in 2021 to the corresponding renewable energy generation from solar (reg(solar)) or wind (reg(wind)). Our intention is not to predict future capacity additions with this proxy, but to illustrate the scale of generation required to close the supply-demand gap under inelastic demand and more granular temporal matching rules. For the sake of simplicity, the proxy assumes that all additional generation also issues GOs. As this is not necessarily the case in practice, the resulting estimates may underestimate the required volumes.

### Reporting summary
Further information on research design is available in the Nature Portfolio Reporting Summary linked to this article.

## Data availability
Source data are provided with this paper. The analysis data is available on GitHub https://github.com/HannaFScholta/Temporal-matching-as-an-accounting-principle-for-green-electricity-claims and has been archived, alongside the code, on Zenodo https://doi.org/10.5281/zenodo.17198701[66]. The raw data used in this work is publicly available

from the cited sources, which are also linked in the GitHub repository. Source data are provided with this paper.

## Code availability
All data preprocessing scripts and the analysis code is available on GitHub https://github.com/HannaFScholta/Temporal-matching-as-an-accounting-principle-for-green-electricity-claims. All code has been archived on Zenodo under https://doi.org/10.5281/zenodo.17198701[66].

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

## Acknowledgements
We thank the Technical University of Munich for its support through the Graduate Program and for providing Open Access funding (Open Access funding enabled and organized via the nation-wide Projekt DEAL agreement with Springer NATURE). We are especially grateful to Prof. Dr. Gunther Friedl for his continuous guidance and feedback, and to our colleagues Dr. Sarah Steinbach and Jeana Ren for their invaluable support, as well as to Prof. Dr. Anders Bjørn and Dr. Lissy Langer (Technical University of Denmark) for their insightful and constructive comments that helped us refine the manuscript. A special thank you also goes to Benedikt Lechl for his exceptional support in retrieving and processing the entso-e data.

## Author contributions
H.F.S. conceived the initial idea and, together with M.J.B., designed the study. H.F.S. conducted the data analyses and coding, wrote the initial draft, and both authors contributed to the final manuscript.

## Funding

## Competing interests
The authors declare no competing interests.
