## [Transparent Peer Review file · Nature Communications]

Temporal matching as an accounting principle for green electricity claims

Corresponding Author: Ms Hanna Scholta

Version 0:

Reviewer comments:

Reviewer #1

(Remarks to the Author)

This paper compares various matching strategies in voluntary renewable energy procurement. The primary insight, in my view, is that a quarterly matching framework could provide a practical and effective compromise that achieves some of the benefits of granular matching without the unjustifiable costs of hourly matching. The modeling exercise is interesting and that insights is valuable, though I raise some issues with the framing.

The authors pursue two conflicting goals without ever addressing that conflict. The authors explore how different matching strategies affect the “sound”-ness of claims and how “to best incentivize additional renewable energy source installations and flexibility measures.” The authors seem to assume these goals go together. They do not. The best way to “incentivize additional renewable energy” is a framework that allows buyers to maximize additional energy per unit of investment regardless of where or when that energy is generated (to abuse the cliché, the climate doesn’t care where or when the energy is generated as long as it reduces emissions). Claim soundness is an entirely distinct goal. Insofar as soundness means the equation of claims with “physical” energy use, it implies serious constraints on where and when energy is generated that will prevent buyers from making the most cost-effective investments. I could, for instance, maximize the soundness of my claim by going off grid using rooftop solar plus storage, but it would be astronomically more expensive than procuring off-site renewables, and society would not be better off as a result.

Of those two goals, the authors focus heavily on the first notion of “sound” claims. Throughout the paper the authors focus on “uncovered intervals” and whether matching provides the “right incentives” to remove those uncovered intervals. The authors should make a case for why soundness matters and then acknowledge the limitations of soundness metrics like uncovered intervals. Just to illustrate the point: if buyer X invests in 100 MW of additional capacity that it can soundly claim to “use,” is that better than buyer Y spending the same amount of money to deploy 200 MW of additional capacity that it doesn’t try to claim to physically “use”? If so, why? If not, what is the value of “soundness” in claims? The answer to that last question is, in my view, best enunciated by the researchers at Princeton, who essentially argue that soundness through granular matching incentivizes investments in a more diverse portfolio. That’s great, but it just begs the question: why not explore measures that directly (rather than indirectly) motivate such investment?

The authors emphasize the importance of “uncovered intervals,” but the case for that metric is not straightforward. Again, just to illustrate the point, if buyer X invests in something like 100 MW of capacity that eliminates all uncovered intervals, is that inherently better than buyer Y investing in 1,000 MW of capacity but having some uncovered intervals? If the cost of reducing uncovered intervals by 10% is reducing deployed capacity by 50%, is that an unambiguous win? The authors should explore these tensions and discuss the limitations of uncovered intervals as a metric of interest.

The Introduction mis-portrays two points. First, the authors state that suppliers “advertise their energy as green” but “cannot guarantee that the physical energy consumed is indeed produced from renewable sources.” This is true but it implies a problem that does not exist. No one can guarantee anything about the “physical energy consumed” anywhere, at any time, on any grid. The second an electron enters the grid it becomes an indistinguishable component of a homogeneous flow. The authors seem to imply some type of false advertising. In reality, suppliers are complying with the laws and market frameworks that dictate renewable energy use claims. Annual matching is a widely-accepted legal fiction. Temporal matching improves on that fiction, but hourly matching is still a fiction. Second, the authors state that renewable energy use

“may also be accounted for by renewable (excess production from another day or another geographical location via energy attribute certificates.” The implication seems to be that EACs are somehow a false form of energy that is different from “physical energy consumed.” There is no such distinction in the law, which is the only dimension that matters here since electricity cannot be physically tracked. Every single legal claim to renewable energy is based on EACs. With the exception of off-grid systems (which we are not talking about), it would make no sense to base on an energy claim on “physical energy consumed.” Such allusions to this dichotomy should be removed.

Last, there are many assertions about the role of EAC and GO prices here. To name just one, the authors state that “the abundant supply of GOs reduces the price premiums to a level too low to incentivize further investments in new capacity.” They support this claim by citing a study based on a single year of data from the Dutch market. I appreciate that such claims are common in the literature, but as a referee I’m going to try to weed them out of this paper. There is no concept in economics that would suggest that a price can be too low to affect supply. Scholars like to focus on exceptional cases (Norwegian hydro RECs being a common and valid one), but as a rule renewable energy development is highly competitive and margins are extremely tight. In that environment there is no logic to the assertion that low prices equate to low impacts on new capacity. Prices are a market output as much as they are a market input and should be described accordingly. If prices are very low, it means an absence of scarcity that could be a signal of a potential market issue (such as the case of Norwegian hydro clearly undermining any assertion of additionality), which is not the same as saying that low-priced EACs cannot support new investment as a rule.

(Remarks on code availability)

I do not have time to replicate the code, but the repository looks well organized, with a clear structure connecting input data to the scripts written in R. At a minimum, the organization suggests that the authors would be willing to help prospective users to replicate the analysis, in the event of any issues in the replication package.

Reviewer #2

(Remarks to the Author)

Key results

European consumers purchase green energy certificates (GECs, in Europe, GOs) to claim emissions reductions based on annual GOs issued and cancelled by the consumers. This is claimed to be neither transparent nor incentivizing emissions reductions. The paper argues for quarterly matching for the short- and hourly matching for the long term to incentivise better demand and supply matching for green energy (GE) and specifically motivating storage solutions.

Validity

There is a relative shortage of quantified studies of the impact of GOs on emissions, as Langer et al. (2023) show, and, as GOs have mostly traded at minimal prices (<€5/MWh compared to electricity prices) and as most European GE is subsidized typically at prices considerably above €50/MWh one would not expect to see much evidence (as is the case). The paper refers to the one case (Xu et al [59]) that finds hourly matching could work. The paper derives data for monthly GO issuance but only has annual data on cancellation. To impute hourly GO issues the paper assumes this to be the same fraction of total GE generated per hour as per month for each GE type (wind, solar in particular). Demand is much cruder – a standard share to total annual demand to total GE supply applied hourly- i.e. demand in each hour for GE is the same fraction of total demand in that hour every hour of the year, so there is no possibility of demand responding to GE supply. The graphs then show that as GE hourly production does not uniformly scale with hourly demand there will be hours of shortage of GE and hours of (relative) surplus. Hence creating an hourly demand for GOs could motivate GE supply to be more responsive (mildly by tuning PV orientation, more by storage of GE to hours of higher GO demand (signalled by higher GO prices). This is wholly unsurprising and the mechanics of generating the data very simple. There is no test (which would almost certainly have to be system-wide modelling, given the low GO prices) of what could be the impact of hourly matching, which other studies have explored and which typically come at a cost that may make this approach more expensive than other policy instruments. As such the paper is fatally flawed.

Significance

The results, given the assumptions, are probably correct but wholly unsurprising, and add nothing to the state of knowledge (see better studies surveyed by Langer et al., 2023 and many of the cited references).

Data and methodology

The authors use GO-data of 2016-2021 with data on European electricity demand and renewable supply to study the quarterly, monthly, weekly, daily, and hourly matching of green electricity supply with demand. This is almost certainly valid but of limited relevance for understanding the likely impact of hourly matching on emissions and costs.

Analytical approach

The approach of looking at past data when GO prices were negligible is unlikely to reverse existing findings of negligible impact while the hourly approach needs system modelling of the type surveyed in Langer et al. (2023) and not undertaken in the paper.

Suggested improvements

None – as similar better work has been published.

Clarity and context

Moderately clearly written although while there is mention of the low GO prices noted by other authors its relevance to the period studied is not emphasised, nor is there any mention of the need for modelling (only one mention to Xu et al’s modelled results).

References

While the list is extensive it omits the key survey article:

Langer, L., M. Brander, S. Lloyd, D. Keles, H. Matthews and A. Bjørn. 2023. Does the purchase of voluntary renewable energy certificates lead to emission reductions? A review of studies quantifying the impact. SSRN at

(Remarks on code availability)

Hardly any data manipulation conducted

Reviewer #3

(Remarks to the Author)

Referee Report Shedding Light on Green Claims: the impact of a closer temporal alignment of supply and demand in voluntary green electricity markets

This paper studies the lack of temporal matching between renewable energy production and consumption in voluntary green electricity markets. Such a mismatch does not provide the right incentives in the sense that certificates that provide time-independent returns curb price signals towards off-peak green electricity generation and reduce the incentives to install flexibility measures such as battery storage systems and demand-side management that would otherwise help balance supply and demand over time.

The mismatching problem has become more salient due to increasing reliance on electricity as a decarbonization strategy and the use of solar energy as an important component of the energy mix. The paper's contribution is to provide a quantification of the mismatch at different granular frequencies for the European guarantees of origin market. The authors argue for an improved match. Even by moving from a yearly match to a quarterly match, we could observe an improvement without significantly increasing the administrative costs of the system.

In general, I think the question is relevant. Nevertheless, my major criticism, concern, and suggestion is that the authors could do more than quantify the mismatch. They could provide some estimates that exemplify the magnitude of the gains of opting for closer matching between generation and demand. Such estimations would provide clear insights about what is really at stake, and how much might we gain if we follow the authors' suggestion.

Further issues that the paper should address:

1) I lack a general discussion about green electricity markets. Voluntary markets often exist alongside regulatory markets and can be influenced by policies that encourage or mandate renewable energy usage. Mandatory commitment to renewable energy can be observed in various regions, including the European Union's (EU) member states. For instance, the new EU's Renewable Energy Directive sets a binding target for the EU to achieve at least 42.5% of its energy consumption from renewable sources by 2030, with an aspiration to reach up to 45%. In this context, how important are voluntary markets, and why is it relevant to study just them? How will the development of mandatory markets affect the need and performance of voluntary markets over time? Moreover, how do mandated markets deal with the mismatch between generation and usage, and what lessons can be drawn from those markets to improve the performance of voluntary markets?

2) Why is the match taking place on an annual basis, and what currently prevents a more granular mismatch? Understanding these constraints is crucial for proposing practical solutions.

3) What is actually at stake if we implement a closer match between demand and supply? I think this is the relevant question that would motivate actual policy action. The analysis so far highlights a problem, but how costly is the problem, and what are the gains of solving it? What should the prices actually look like, and what type of incentives would this bring about? I encourage the authors to provide such calculations to enhance the novelty of the analysis and increase the policy relevance of its conclusions.

4) In the context of the European Union, the paper could explore how closer temporal matching could impact market integration and cross-border electricity trade. Given the interconnected nature of the EU's electricity markets, the alignment of supply and demand across borders could be a critical factor in achieving a more efficient and resilient energy system. The potential for cross-border cooperation and the harmonization of temporal matching standards could also be discussed.

(Remarks on code availability)

Version 1:

Reviewer comments:

Reviewer #1

(Remarks to the Author)

This paper provides many interesting insights. I think it could make a valuable contribution worthy of Nature Communications. But the authors still make many assertions and implied conclusions that are not supported by the evidence presented here.

In my initial comments I distinguished between "soundness" and the broad concept of real-world impact. The authors

analyze here a framework on soundness and provide some evidence for how different proposals achieve different degrees of soundness (as defined by the authors). That is fine and useful. But the authors didn't heed my previous points that there is no scientific connection between soundness and real-world impact, though that connection is implied throughout the paper.

I didn't need to read beyond the abstract to see that. The authors state in the second sentence that critics argue that certificates do not incentivize renewable deployment, and the third sentence states that EU legislation "calls for transparent and reliable green claims." The implication is that "transparent and reliable green claims" provide a solution for the problem stated in the prior sentence. That is a non-sequitur. The "transparency and reliability" of claims will vary based on whatever set of accounting principles we decide to apply. That choice is subjective, as the authors themselves clearly state on page 9: "According to our beliefs, green electricity claims must accurately reflect this principle to be considered transparent and reliable." Fair enough, those are your beliefs, and you could write an interesting paper exploring how to achieve "transparency and reliability" according to your beliefs. But where is the connection between your beliefs and real-world impact?

The authors made some worthy attempts to heed my comments, but the implied connection between soundness and impact remains largely intact in the revised manuscript. Broad swathes of the intro remain dedicated to critiques of impact—not soundness. I see that conversation as largely irrelevant to the analysis, but the implication is that the ensuing analysis provides some sort of solution to critiques of impact.

The authors make the connection more concrete in the conclusion, where they state for instance that their principles are needed "to allow voluntary EAC markets to steer investments toward the physically required measures." The authors never offer any scientific principle to make the connection. The closest they come is a speculation on page 9 that "Such a shift could incentivize an expansion of renewable capacity." It could, but it also certainly could not, and indeed could do exactly the opposite if these proposals increase costs so much that voluntary buyers stop voluntarily procuring renewable energy.

My fundamental point is that the authors go to great and unnecessary lengths to criticize existing frameworks based on their real-world impacts. But this is not a paper about impact, it is a paper about the information-gathering exercise of emissions accounting. I encourage the authors to sharpen the focus of this paper by removing the superfluous background discussion and unfounded conclusions about the implications of soundness for impact.

Further, the authors should be very clear that "soundness" is defined in a very specific way for the purposes of this paper. For instance, if I can scientifically demonstrate that my investment in a solar plant in a foreign country reduces emissions exactly equal to my own emissions, why isn't that "sound"? Many stakeholders in this discourse equate "soundness" with an accurate matching of emissions generated to emission reduced. The authors define it otherwise according to their own "beliefs", and that is fine, but it should be clear. It would be very clear if they reminded the reader through with terms such as "soundness under temporal matching principles" or something to that effect.

Some more minor points:

- The authors in a few spots refer to green claims based on "excess" production. Excess to what? If the power is being claimed in RECs that means the power was used, it wasn't excess to the system, it was excess to the buyer's profile, which means the power is only "excess" as defined in your very particular framework.
- The authors took a step in the right direction with the reframed focusing question "Would green electricity claims still stand under stricter temporal matching?" But the question is loaded. We have no reason to suspect that claims would "still stand" if we change accounting requirements. The question should how new principles would affect compliance, not if.
- The authors wordsmithed some points about the relationship between price and impact, but maintained the message with the revised language "revenues... that only provide marginal incentives for further investments in new capacity." It is fundamentally the same point and encounters the same problems. There is nothing in economic theory to explain why additional revenues wouldn't affect supply in a highly-competitive industry like renewable energy development. Those effects are, of course, "marginal", but who decides how marginal is "enough"? In line with my comments above, I think the paper would be stronger by omitting this whole line of reasoning.
- Just as an example of why the background discussion weakens the paper, consider this line: "the systematic failure to consider the actual physical flows of electricity..." Each REC represents 1 MWh of renewable electricity and its attributes as metered and "physically" delivered instantaneously on the grid where it was generated. Was that electricity not "physical"? If "actual physical flow" implies the flow of electricity "from" a generator "to" the end user then the only way to "consider the actual physical flows of electricity" is to disconnect from the grid and only claim use off of an islanded system. Again, as I mentioned in my last round of comments, I realize that this sort of rhetoric is common, but as a referee I want good research like this paper to be published without these straw-man arguments.
- There are a few spots where the authors discuss the voluntary market in compliance terms, most notably in the conclusion where the authors ask "whether and which penalties should occur in case of non-compliance." No such penalties exist, this is a voluntary market, and that is fundamentally the problem. We can ask corporations to report in certain ways, maybe we can even require certain accounting practices, but we cannot require corporations to change their operations according to those practices.

To summarize, I like the insights this paper provides and I would like to see it published, but I can't endorse this paper as is given the significant overreach of the assertions beyond the scientific evidence presented in the paper.

(Remarks on code availability)

n/a

Reviewer #3

(Remarks to the Author)

The authors have addressed all of my comments satisfactorily. I believe the paper has improved significantly, particularly in clarifying the relevance and implications of the lack of temporal matching.

(Remarks on code availability)

I could not access the link to the code.

Version 2:

Reviewer comments:

Reviewer #1

(Remarks to the Author)

I thank the authors for taking my comments seriously and taking the time to address my points in their revisions. As the reviewers can probably tell, this is an important topic to me, and an area where I feel that mischaracterizations have been perpetuated in the literature. I remain skeptical of the implied connection between "soundness" and impact, but the authors have sufficiently downplayed the connection to the point that it no longer poses a fundamental issue for the paper, in my view. The study offers interesting insights and a useful summary of the debate, in general.

If there is time to address one suggestion, I would still recommend to define "soundness." Again, I don't think there is anything inherently unsound about annual matching. Any accounting approach only becomes unsound if we decide we want to use it for a different purpose. The annual rainfall of a country, for instance, may be a sound way of describe how rainy a country is in aggregate, but an unsound way of predicting how likely it is to rain on a specific day. Soundness, in this study, implies a view of accuracy based on physical electricity flows. It's fine to explore that view, but it is not universal, and available research does not support the premise that that specific view of soundness will enhance the impact of emissions accounting exercises.

(Remarks on code availability)

Response to Reviewer #1

Dear reviewer,

we would like to thank you for your valuable feedback and comments we received. These comments were very helpful and gave us the chance to fully rework and improve our paper with clearer and more precise formulations as well as more detailed and nuanced discussions. In the following, we will state the changes we introduced to reflect your guidance.

This paper compares various matching strategies in voluntary renewable energy procurement. The primary insight, in my view, is that a quarterly matching framework could provide a practical and effective compromise that achieves some of the benefits of granular matching without the unjustifiable costs of hourly matching. The modeling exercise is interesting and that insights is valuable, though I raise some issues with the framing.

The authors pursue two conflicting goals without ever addressing that conflict. The authors explore how different matching strategies affect the “sound”-ness of claims and how “to best incentivize additional renewable energy source installations and flexibility measures.” The authors seem to assume these goals go together. They do not. The best way to “incentivize additional renewable energy” is a framework that allows buyers to maximize additional energy per unit of investment regardless of where or when that energy is generated (to abuse the cliché, the climate doesn’t care where or when the energy is generated as long as it reduces emissions). Claim soundness is an entirely distinct goal. Insofar as soundness means the equation of claims with “physical” energy use, it implies serious constraints on where and when energy is generated that will prevent buyers from making the most cost-effective investments. I could, for instance, maximize the soundness of my claim by going off grid using rooftop solar plus storage, but it would be astronomically more expensive than procuring off-site renewables, and society would not be better off as a result.

Of those two goals, the authors focus heavily on the first notion of “sound” claims. Throughout the paper the authors focus on “uncovered intervals” and whether matching provides the “right incentives” to remove those uncovered intervals. The authors should make a case for why soundness matters and then acknowledge the limitations of soundness metrics like uncovered intervals. Just to illustrate the point: if buyer X invests in 100 MW of additional capacity that it can soundly claim to “use,” is that better than buyer Y spending the same amount of money to deploy 200 MW of additional capacity that it doesn’t try to claim to physically “use”? If so, why? If not, what is the value of “soundness” in claims? The answer to that last question is, in my view, best enunciated by the researchers at Princeton, who essentially argue that soundness through granular matching incentivizes investments in a more diverse portfolio. That’s great, but it just begs the question: why not explore measures that directly (rather than indirectly) motivate such investment?

Thank you for pointing out this important issue and providing such a detailed elaboration on the discrepancies in goals! We fully agree. It is essential to clarify the goals and address the potential conflicts of soundness and how to incentivize additional renewable energy. We have changed several aspects within the paper to address your points and clarify the conflict and the interplay between soundness and additionality:

- We have clarified that the focus of our paper is on claim soundness, as reflected in our updated research questions: “Would green electricity claims still stand under stricter temporal matching? Which effects may be expected from different temporal matching frequencies?” Additionally, we have expanded on the importance of claim soundness, highlighting how it enables consumers to take meaningful action against climate change. To strengthen this argument, we have incorporated references to relevant EU policy frameworks, such as the Green Industrial Plan and legislative proposals like the Green Claims Directive.
- We have further elaborated on the interplay between sound claims and additional renewable energy, emphasizing that sound claims should meaningfully support the development of a resilient renewable energy system – one that reliably meets demand at all times. We have

clarified in our discussion and introduction sections, that effectively combating climate change requires not just incentivizing additional renewable generation but also prioritizing the “right” kind – such as west- or east-facing photovoltaic installations and flexibility measures like storage. We now better elaborate and discuss the value of sound claims as lying in the ability to align investments with system-level needs, promoting the development of renewable energy in a way that supports grid reliability and resilience. We have highlighted this point further in our paper to provide additional clarity and context and clarify the questions you have raised with your example

The authors emphasize the importance of “uncovered intervals,” but the case for that metric is not straightforward. Again, just to illustrate the point, if buyer X invests in something like 100 MW of capacity that eliminates all uncovered intervals, is that inherently better than buyer Y investing in 1,000 MW of capacity but having some uncovered intervals? If the cost of reducing uncovered intervals by 10% is reducing deployed capacity by 50%, is that an unambiguous win? The authors should explore these tensions and discuss the limitations of uncovered intervals as a metric of interest.

We sincerely thank you for this thoughtful comment regarding the limitations and implications of focusing on “uncovered intervals” as a metric. To address this comment, we now include a way more elaborated discussion on claim soundness to the broader goal of building a resilient renewable energy system – one capable of reliably meeting demand at all times. This framing provides the reader with a clearer understanding of the metric’s relevance. However, we also acknowledge its limitations, particularly when considered in isolation.

To address this further, we have extended our analysis to include an additional metric, which evaluates the volumetric mismatches between renewable supply and demand. Chapter 3.1 now focuses on our findings related to the “uncovered intervals” metric, while Chapter 3.2 complements these findings by confirming the previously identified trends through the new metric “volumetric scale of under-coverage” and providing further insights into the implications of stricter temporal matching.

Additionally, we have revised the conclusion to acknowledge that the flexibility allowed within the system – for instance, the conditions under which mismatches may be tolerated and the potential penalties for non-compliance – are important to consider.

The Introduction mis-portrays two points. First, the authors state that suppliers “advertise their energy as green” but “cannot guarantee that the physical energy consumed is indeed produced from renewable sources.” This is true but it implies a problem that does not exist. No one can guarantee anything about the “physical energy consumed” anywhere, at any time, on any grid. The second an electron enters the grid it becomes an indistinguishable component of a homogeneous flow. The authors seem to imply some type of false advertising. In reality, suppliers are complying with the laws and market frameworks that dictate renewable energy use claims. Annual matching is a widely-accepted legal fiction. Temporal matching improves on that fiction, but hourly matching is still a fiction. Second, the authors state that renewable energy use “may also be accounted for by renewable (excess production from another day or another geographical location via energy attribute certificates.” The implication seems to be that EACs are somehow a false form of energy that is different from “physical energy consumed.” There is no such distinction in the law, which is the only dimension that matters here since electricity cannot be physically tracked. Every single legal claim to renewable energy is based on EACs. With the exception of off-grid systems (which we are not talking about), it would make no sense to base on an energy claim on “physical energy consumed.” Such allusions to this dichotomy should be removed.

Thank you for bringing these misrepresentations to our attention. We fully agree that the formulations may cause misinterpretations and may imply that suppliers are breaking the law or failing to comply with market frameworks. Following your guidance, we have revised this section as follows:

“As electricity is a homogeneous good, the green labeling of energy requires the cancellation of energy attribute certificates [...]. To make a green claim, providers typically have to demonstrate that an equivalent volume of renewable energy has been produced over the course of the respective year [...].

This approach, commonly referred to as annual volumetric matching, allows green claims to be based on renewable (excess) production from a time and location different than that of consumption, as closer temporal and locational alignment between green energy supply and demand is currently not mandated by legislation and relevant standards.”

Additionally, we clarified all other similar formulations as of our previous draft to avoid any misrepresentations. This revision should ensure that readers are not misled into thinking that suppliers are engaging in improper practices or that there would be any claims that are based on “physical energy consumed”.

Last, there are many assertions about the role of EAC and GO prices here. To name just one, the authors state that “the abundant supply of GOs reduces the price premiums to a level too low to incentivize further investments in new capacity.” They support this claim by citing a study based on a single year of data from the Dutch market. I appreciate that such claims are common in the literature, but as a referee I’m going to try to weed them out of this paper. There is no concept in economics that would suggest that a price can be too low to affect supply. Scholars like to focus on exceptional cases (Norwegian hydro RECs being a common and valid one), but as a rule renewable energy development is highly competitive and margins are extremely tight. In that environment there is no logic to the assertion that low prices equate to low impacts on new capacity. Prices are a market output as much as they are a market input and should be described accordingly. If prices are very low, it means an absence of scarcity that could be a signal of a potential market issue (such as the case of Norwegian hydro clearly undermining any assertion of additionality), which is not the same as saying that low-priced EACs cannot support new investment as a rule.

Thank you for this valuable recommendation! We fully agree that precise wording is crucial, even if other literature sometimes addresses these topics more loosely. Following your suggestion, we have removed any statements implying that low-priced EACs categorically cannot support new investments.

Instead, we have reframed the relevant sections to focus on (abundant) supply and the resulting (lack of) scarcity, as you suggested. For example, when referring to the study on the Dutch market, we now state: “The abundant supply of GOs reduces scarcity, driving revenues to levels that only provide marginal incentives for further investments in new capacity.” Additionally, we have revised the language throughout our manuscript to ensure it aligns more closely with market economics.

-

At this stage, we would like to express our gratitude for your detailed and valuable feedback that clearly improved the framing and wording as well as the discussion parts of this paper! We hope our revisions have addressed your comments to your satisfaction.

Response to Reviewer #2

Dear reviewer,

we appreciate the time you invested in providing a thorough and critical review of our manuscript. Your comments have challenged us to re-evaluate, clarify, and strengthen the contributions of our work. We believe these revisions have significantly improved the quality and clarity of our research. Below, we address each of your points and outline the changes we have made in response.

Key results

European consumers purchase green energy certificates (GECs, in Europe, GOs) to claim emissions reductions based on annual GOs issued and cancelled by the consumers. This is claimed to be neither transparent nor incentivizing emissions reductions. The paper argues for quarterly matching for the short- and hourly matching for the long term to incentivise better demand and supply matching for green energy (GE) and specifically motivating storage solutions.

Validity

There is a relative shortage of quantified studies of the impact of GOs on emissions, as Langer et al. (2023) show, and, as GOs have mostly traded at minimal prices (<€5/MWh compared to electricity prices) and as most European GE is subsidized typically at prices considerably above €50/MWh one would not expect to see much evidence (as is the case). The paper refers to the one case (Xu et al [59]) that finds hourly matching could work. The paper derives data for monthly GO issuance but only has annual data on cancellation. To impute hourly GO issues the paper assumes this to be the same fraction of total GE generated per hour as per month for each GE type (wind, solar in particular). Demand is much cruder – a standard share to total annual demand to total GE supply applied hourly- i.e. demand in each hour for GE is the same fraction of total demand in that hour every hour of the year, so there is no possibility of demand responding to GE supply. The graphs then show that as GE hourly production does not uniformly scale with hourly demand there will be hours of shortage of GE and hours of (relative) surplus. Hence creating an hourly demand for GOs could motivate GE supply to be more responsive (mildly by tuning PV orientation, more by storage of GE to hours of higher GO demand (signalled by higher GO prices). This is wholly unsurprising and the mechanics of generating the data very simple. There is no test (which would almost certainly have to be system-wide modelling, given the low GO prices) of what could be the impact of hourly matching, which other studies have explored and which typically come at a cost that may make this approach more expensive than other policy instruments. As such the paper is fatally flawed.

Significance

The results, given the assumptions, are probably correct but wholly unsurprising, and add nothing to the state of knowledge (see better studies surveyed by Langer et al., 2023 and many of the cited references).

Thank you for your pointed critique on the validity and significance of our manuscript and analysis. While some of our results may appear unsurprising from a technical perspective, we argue that they remain highly relevant, amongst others, in light of the upcoming 2025 review of the European Guarantees of Origin (GO) system, which seeks to evaluate the balance of supply and demand within the GO market. Following your critical but valuable feedback, our revised paper from the introduction onwards now better communicates its key proposition, which is to offer an accounting perspective on green electricity claims under shorter-than-annual matching.

Further improving on our acknowledgment of the cost implications of hourly matching, we have expanded the literature review in the introduction to include additional key modeling studies as starting point of our analysis, such as Riepin & Brown (2024) and those reviewed by Langer et al. (2024). We have revised our manuscript to clearly state how our analysis complements these works with an accounting perspective grounded in empirical data. This perspective highlights how claim soundness could already benefit from quarterly or monthly matching, which can be implemented within the already existing certificate schemes using monthly production time stamps to address seasonal imbalances. We

further contribute a detailed discussion on how transitioning to hourly matching in the long term could significantly enhance the transparency and reliability of green electricity claims by addressing day-night disparities and providing incentives for investments in off-peak renewable generation and storage solutions. Besides having revised our manuscript to more clearly articulate this contribution, we have additionally provided more room to our primary findings, e.g. by moving the text passages and figure on the hypothetical scenarios, which served solely to clarify the previously outlined dynamics, to the appendix.

Thanks to your comments, we have now clarified our contribution, elaborated more on the empirical insights and provide a stronger discussion on policy implications for refining EAC schemes in the short and long term, thus complementing the findings of previous modeling studies.

Data and methodology

The authors use GO-data of 2016-2021 with data on European electricity demand and renewable supply to study the quarterly, monthly, weekly, daily, and hourly matching of green electricity supply with demand. This is almost certainly valid but of limited relevance for understanding the likely impact of hourly matching on emissions and costs.

Thank you for your feedback and the chance to clarify this for the reader. We now provide a clearer introduction to our analysis, highlighting that our analysis leverages empirical data from 2016–2021 to provide an accounting perspective on the temporal alignment of green electricity supply and demand. We have further emphasized that the identification of existing temporal imbalances yields insights into the soundness and implications of green claims within the current EAC systems. While we fully recognize the importance of comprehensive modeling studies to evaluate the impact of hourly matching on emissions and costs, our findings complement these studies by providing valuable insights into green electricity claims and the underlying dynamics at play from an accounting perspective.

We have further revised and expanded the results section to, within the scope of our empirical methodology, explore the potential implications of shifting to more granular matching periods. Specifically, in the newly developed Chapter 3.2, we estimate the scale-up in renewable energy generation required to address under-coverages. This includes quantifying the additional solar and/or wind generation needed to cover the most undersupplied intervals under each matching scheme analyzed. Although it does not achieve the comprehensive depth of system modeling, this analysis provides valuable insights into the scale of changes in renewable energy generation driven by stricter temporal matching.

Analytical approach

The approach of looking at past data when GO prices were negligible is unlikely to reverse existing findings of negligible impact while the hourly approach needs system modelling of the type surveyed in Langer at al. (2023) and not undertaken in the paper.

Suggested improvements

None – as similar better work has been published.

Clarity and context

Moderately clearly written although while there is mention of the low GO prices noted by other authors its relevance to the period studied is not emphasised, nor is there any mention of the need for modelling (only one mention to Xu et al's modelled results).

Thank you for raising these concerns and for making us aware that our original formulation may have inadvertently given rise to these impressions. We appreciate the opportunity to clarify our contributions more effectively and have revised the paper as follows:

- We have completely rewritten the introduction and conclusion capter and further clarified that the low GO prices do not diminish the validity of our results as we find current market rules to

not provide the right incentive mechanisms – even with potential increases in GO prices. The underlying mechanics our analysis reveals – such as the persistent temporal misalignment between green electricity supply and demand – are intrinsic to the current design of the GO system and remain valid even if GO prices increase significantly. We have strengthened our focus on highlighting these structural limitations and advocating for improvements to the system, irrespective of market price levels.

- Additionally, to address your concerns, we have expanded the paper to give greater attention and appreciation to related modeling studies beyond Xu et al. (2024), such as those by Riepin & Brown (2024) and the review of modeling studies by Langer et al. (2024). These studies examine emissions and cost implications, and we position our work as a complementary contribution, offering an empirical accounting perspective that focuses on the limitations of the current EAC frameworks and potential refinements. By doing so, we aim to enrich the broader discussion on granular matching and its role in improving green electricity claims.

References

While the list is extensive it omits the key survey article:

Langer, L., M. Brander, S. Lloyd, D. Keles, H. Matthews and A. Bjørn. 2023. Does the purchase of voluntary renewable energy certificates lead to emission reductions? A review of studies quantifying the impact. SSRN at https://papers.ssrn.com/sol3/papers.cfm?abstract_id=4636218

Thank you very much for suggesting this article. As it was recently published in the Journal of Cleaner Production, we included the published version of it in our revised manuscript.

-

At this stage, we would like to express our gratitude for your sharp and critical assessment that has clearly helped us improve our manuscript. We hope our revisions have addressed your comments to your satisfaction.

Response to Reviewer #3

Dear reviewer,

we would like to thank you for your valuable feedback and comments, which have allowed us to significantly improve our work. In the following, we will state the changes we introduced to reflect your guidance.

This paper studies the lack of temporal matching between renewable energy production and consumption in voluntary green electricity markets. Such a mismatch does not provide the right incentives in the sense that certificates that provide time-independent returns curb price signals towards off-peak green electricity generation and reduce the incentives to install flexibility measures such as battery storage systems and demand-side management that would otherwise help balance supply and demand over time.

The mismatching problem has become more salient due to increasing reliance on electricity as a decarbonization strategy and the use of solar energy as an important component of the energy mix. The paper's contribution is to provide a quantification of the mismatch at different granular frequencies for the European guarantees of origin market. The authors argue for an improved match. Even by moving from a yearly match to a quarterly match, we could observe an improvement without significantly increasing the administrative costs of the system.

In general, I think the question is relevant. Nevertheless, my major criticism, concern, and suggestion is that the authors could do more than quantify the mismatch. They could provide some estimates that exemplify the magnitude of the gains of opting for closer matching between generation and demand. Such estimations would provide clear insights about what is really at stake, and how much might we gain if we follow the authors' suggestion.

Thank you for your thoughtful and constructive feedback. We fully agree that providing estimations about the potential gains from closer matching significantly enhances the practical relevance and depth of our analysis. We addressed your comment and now provide estimates on the impact of stricter temporal matching on the scale-up of renewable energy generation. We have, therefore, restructured our Results and Discussion section as follows:

Chapter 3.1 now includes all our previous findings. The newly developed Chapter 3.2 focuses on the implications of stricter temporal matching schemes and presents our estimates of the renewable energy generation scale-up needed to address under-coverages. Specifically, we quantify the additional solar and/or wind generation required to cover the most undersupplied intervals under each matching scheme - grounding our focus on solar and wind energy in the EU's expansion plans for renewable generation.

Key findings include:

- **Quarterly Matching:** Resolving the under-coverage of the most undersupplied interval would require either a 5% increase in wind generation, a 30% increase in solar generation, or a combination of both.
- **Hourly Matching:** Under-coverage cannot be addressed through solar generation. Achieving balance would, therefore, require up to a 211% increase in wind generation.

These estimates highlight the substantial scale of additional renewable generation required to meet the demands of stricter matching requirements. While this section focuses on generation implications, we emphasize in the discussion that also other solutions – such as holistically integrating energy storage into the EAC value chain and managing demand – are critical to mitigating shortfalls.

Besides restructuring the Results and Discussion section, we have also expanded the Methods section to provide a clear explanation of how the estimates were calculated.

Further issues that the paper should address:

1) I lack a general discussion about green electricity markets. Voluntary markets often exist alongside regulatory markets and can be influenced by policies that encourage or mandate renewable energy usage. Mandatory commitment to renewable energy can be observed in various regions, including the European Union's (EU) member states. For instance, the new EU's Renewable Energy Directive sets a binding target for the EU to achieve at least 42.5% of its energy consumption from renewable sources by 2030, with an aspiration to reach up to 45%. In this context, how important are voluntary markets, and why is it relevant to study just them? How will the development of mandatory markets affect the need and performance of voluntary markets over time? Moreover, how do mandated markets deal with the mismatch between generation and usage, and what lessons can be drawn from those markets to improve the performance of voluntary markets?

Thank you for this suggestion to provide clarifications on the broader context of green electricity markets. We have strengthened our reasoning for focusing on voluntary markets, grounding it in the emphasis placed by the EU and other entities on the role of consumers in combating climate change. To substantiate this argument, we have incorporated references to current EU policy frameworks, including the Green Industrial Plan and legislative proposals such as the Green Claims Directive.

Additionally, we have elaborated on the interplay between voluntary and mandatory certificate markets, providing further clarification on what mandatory certificate markets are:

- “Such voluntary consumption of energy that is claimed to be green can complement governmental policies aimed at promoting renewable energy adoption, such as feed-in tariffs, tax credits, or mandates for green energy usage. In contrast to the voluntary demand for green energy, demand in mandated markets arises from government-enforced annual quota obligations (partially referred to as renewable portfolio standards). Non-compliance with quotas typically requires penalty payments. See e.g. the Swedish-Norwegian Tradable Green Certificate Market and Renewable Energy Certificate (REC) compliance markets in the United States.”

We have also addressed why mandatory certificate markets have so far not faced structural issues like oversupply:

- “In mandatory green electricity markets, where demand and supply are regulated through government-imposed quotas adjusted over time, structural oversupply has not similarly mitigated scarcity.”

Finally, while our analysis focuses on voluntary markets, we acknowledge that the findings could hold additional relevance for improving mandatory markets in the future. Specifically, they could help shift the focus in these markets from incentivizing additional megawatts of generation to encouraging, for instance, off-peak solar generation:

- “While our analysis centers on voluntary green electricity markets, these findings may also provide relevant information for future enhancements of and incentivization in mandatory markets.”

2) Why is the match taking place on an annual basis, and what currently prevents a more granular mismatch? Understanding these constraints is crucial for proposing practical solutions.

Thank you for highlighting the need to address these points more thoroughly. We have clarified that the predominant use of annual matching stems from the absence of legislation and standards mandating more granular approaches.

In addition, we have expanded on the constraints preventing more granular matching by discussing the limitations of existing EAC schemes. Specifically, we note that substantial modifications, such as incorporating more precise timestamps, would be required for these schemes to allow for granular approaches like hourly matching.

3) *What is actually at stake if we implement a closer match between demand and supply? I think this is the relevant question that would motivate actual policy action. The analysis so far highlights a problem, but how costly is the problem, and what are the gains of solving it? What should the prices actually look like, and what type of incentives would this bring about? I encourage the authors to provide such calculations to enhance the novelty of the analysis and increase the policy relevance of its conclusions.*

We truly appreciate you emphasizing that the costs and gains of closer temporal matching are highly interesting. While primarily aiming to provide an accounting perspective on green claims and closer temporal matching based on empirical data, we have enhanced our considerations of the cost and gains dimensions. As a proxy for system costs, we have expanded our discussion on the feasibility of different matching schemes, with a particular emphasis on the modifications needed in current systems to enable more granular temporal alignment. We have further strengthened our focus on the benefits of addressing temporal mismatches by dedicating Chapter 3.2 to calculations on the wind and solar scale-ups required to support more granular temporal alignment. Regarding the incentives stricter temporal matching would create, we have further refined our reasoning, streamlining the structure of our arguments to present a clearer and more cohesive case for how stricter temporal matching could encourage investments in renewable generation, energy storage, and demand-side flexibility.

This allows the reader to gain a clearer understanding of what is at stake when transitioning from annual to quarterly, more generally, towards stricter temporal matching. We hope these additions enhance the policy relevance and practical implications of our analysis.

4) *In the context of the European Union, the paper could explore how closer temporal matching could impact market integration and cross-border electricity trade. Given the interconnected nature of the EU's electricity markets, the alignment of supply and demand across borders could be a critical factor in achieving a more efficient and resilient energy system. The potential for cross-border cooperation and the harmonization of temporal matching standards could also be discussed.*

Thank you for your insightful remark regarding the potential impact of closer temporal matching on market integration and cross-border electricity trade within the European Union.

While *locational matching*, defined as the closer alignment of the location of green electricity demand and supply, would significantly restrict cross-border electricity trade, *temporal matching* offers the distinct advantage of preserving the functionality of the European EAC market. This allows for continued cross-border trade while enhancing the alignment of supply and demand over time. We recognize that we had not sufficiently highlighted this benefit of temporal matching in our initial manuscript and have thus revised and extended our introduction. We believe this revision brings greater clarity to the value of temporal matching by ensuring that readers fully understand its impact on cross-border relations and the preservation of trade within multi-country markets.

-

At this stage, we would like to express our gratitude for your detailed and valuable feedback that clearly helped us improve our paper. We hope our revisions have addressed your comments to your satisfaction.

Letter to Reviewer 1: Point-by-Point Response by the Authors

This paper provides many interesting insights. I think it could make a valuable contribution worthy of Nature Communications. But the authors still make many assertions and implied conclusions that are not supported by the evidence presented here. In my initial comments I distinguished between “soundness” and the broad concept of real-world impact. The authors analyze here a framework on soundness and provide some evidence for how different proposals achieve different degrees of soundness (as defined by the authors). That is fine and useful. But the authors didn’t heed my previous points that there is no scientific connection between soundness and real-world impact, though that connection is implied throughout the paper.	We sincerely thank you for your thorough and constructive feedback. It greatly helped us identify and better understand the concerns you raised in your previous review and guided us in revising the manuscript accordingly. In response, we have substantially rewritten the abstract, introduction, conclusion, and relevant parts of the results/discussion section. These revisions aim to sharpen the focus of the paper on its core contribution, namely a framework for evaluating the temporal soundness of green electricity claims, and to remove unfounded assertions regarding the implications of soundness for real-world impact. We believe the paper has been significantly strengthened through the integration of your comments, and we highly appreciate the opportunity to clarify and refine our arguments. Below, we provide a detailed, point-by-point response outlining how we have addressed each of your comments in the revised manuscript. To improve clarity and transparency, we have assigned a brief title to each comment, shown in blue, that summarizes the core of your feedback as we understood it. These captions are intended to ensure that our interpretation is accurate and that our responses directly and explicitly address your concerns.
Comment 1: Refocus the manuscript on temporal claim soundness and remove unsupported links to real-world impact	
I didn’t need to read beyond the abstract to see that. The authors state in the second sentence that critics argue that certificates do not incentivize renewable deployment, and the third sentence states that EU legislation “calls for transparent and reliable green claims.” The implication is that “transparent and reliable green claims” provide a solution for the problem stated in the prior sentence. That is a non-sequitur. The “transparency and reliability” of claims will vary based on whatever set of accounting principles we decide to apply. That choice is subjective, as the authors themselves clearly state on page 9: “According to our beliefs, green electricity claims must accurately reflect this principle to be considered transparent and reliable.” Fair enough, those are your beliefs, and you could write an interesting paper exploring how to achieve “transparency and reliability” according to your beliefs. But where is the connection	Thank you for this important and detailed feedback. We now better understand how elements of the original manuscript - especially the abstract, introduction, and conclusion - could be read as implying a causal or scientific link between claim soundness (as defined by temporal matching) and real-world impact, particularly regarding renewable deployment. We appreciate you pointing to specific instances where this implication arose. To address this, we have made substantial revisions throughout the manuscript: Abstract: We have restructured and rewritten the abstract to avoid implying that transparent and reliable green claims solve the issue of insufficient incentives for renewable deployment. The abstract now focuses on the core contribution of the paper: assessing the temporal soundness of green electricity claims under different matching principles. Introduction and Literature Review: We have substantially revised and shortened this

between your beliefs and real-world impact?

The authors made some worthy attempts to heed my comments, but the implied connection between soundness and impact remains largely intact in the revised manuscript. Broad swathes of the intro remain dedicated to critiques of impact—not soundness. I see that conversation as largely irrelevant to the analysis, but the implication is that the ensuing analysis provides some sort of solution to critiques of impact.

The authors make the connection more concrete in the conclusion, where they state for instance that their principles are needed “to allow voluntary EAC markets to steer investments toward the physically required measures.” The authors never offer any scientific principle to make the connection. The closest they come is a speculation on page 9 that “Such a shift could incentivize an expansion of renewable capacity.” It could, but it also certainly could not, and indeed could do exactly the opposite if these proposals increase costs so much that voluntary buyers stop voluntarily procuring renewable energy.

My fundamental point is that the authors go to great and unnecessary lengths to criticize existing frameworks based on their real-world impacts. But this is not a paper about impact, it is a paper about the information-gathering exercise of emissions accounting. I encourage the authors to sharpen the focus of this paper by removing the superfluous background discussion and unfounded conclusions about the implications of soundness for impact.

section to sharpen the focus on our core contribution. In the previous version, the introduction included summaries of critiques of existing systems that focused on their limited real-world impact. This may have given the impression that our paper aimed to address those impact-related issues directly. Therefore, we have carefully rephrased this section to avoid such implications.

Explicitly stating that much of the existing work on voluntary green electricity markets has taken an impact-oriented perspective, we now position our work in contrast to this literature in the revised version – acknowledging that our paper does not evaluate real-world impacts. We, instead, highlight that we apply temporal matching as an accounting metric to study the soundness of green claims. To further reinforce this focus, we have also clarified and formalized our definition of soundness and now communicate it early in the introduction (see our response to Comment 2).

“So far, voluntary green electricity markets have largely been assessed through the lens of environmental impact: [...]. Similarly, research on temporal matching has so far also focused on environmental impact: [...]. While prior work on temporal matching has focused on system-level impacts, to the best of our knowledge, past literature has not empirically assessed how stricter temporal matching affects the transparency and reliability of green electricity claims. [...] Our paper introduces an accounting perspective to the evolving discussion on granular matching and gives policy recommendations for temporally sound claims, particularly in the context of the current revision of the GHG Scope 2 Guidance”

Discussion and Conclusion:

We have revised the results and initial discussion to focus strictly on what can be inferred from our results and have removed unfounded statements about investment effects or renewable deployment. Where we do discuss potential further implications, we now clearly frame them as hypothetical and highly dependent on market behavior, such as demand elasticity in voluntary markets. For example, we now explicitly note that the uptake of stricter matching schemes could decrease demand due to higher costs, depending on how buyers respond. We do not make claims about which investments would be “steered” by our proposed principles, but instead repeatedly emphasize the voluntary nature of the market and the uncertainty surrounding how actors may respond. In particular, we note that understanding potential real-world effects would require, among other things, further research into the elasticity of demand for voluntary procurement under stricter temporal matching schemes.

	Throughout the manuscript, we have aimed to avoid conflating “temporal soundness” with broader notions of impact and have added clarifying language (e.g., “soundness under temporal matching principles”) to ensure our use of the term remains specific and consistent with the framework we develop (please also see our response to Comment 2 below).
Comment 2: Clarify that ‘soundness’ is defined specifically within the temporal matching framework and avoid ambiguous terminology	
Further, the authors should be very clear that “soundness” is defined in a very specific way for the purposes of this paper. For instance, if I can scientifically demonstrate that my investment in a solar plant in a foreign country reduces emissions exactly equal to my own emissions, why isn’t that “sound”? Many stakeholders in this discourse equate “soundness” with an accurate matching of emissions generated to emission reduced. The authors define it otherwise according to their own “beliefs”, and that is fine, but it should be clear. It would be very clear if they reminded the reader through with terms such as “soundness under temporal matching principles” or something to that effect.	Thank you for highlighting the need to clearly define the specific meaning of “soundness” as used in our paper. We have carefully revised the manuscript to ensure that our use of the term is consistently framed within the context of temporal matching principles, rather than implying a universal or generally accepted definition. Therefore, we have added an explicit definitional paragraph in the Introduction that sets the scope of our analysis: “What is considered transparent and reliable depends on the applied accounting principles. Given that global decarbonization efforts aim to ultimately align total energy demand with renewable supply at all times, we adopt temporal matching as the central principle shaping our definition of claim soundness.” We further explain that within this framing, “sound claims should reflect a consistent temporal alignment between supply and demand.” This added section establishes temporal matching as the specific accounting lens we use throughout the paper and clearly distinguishes it from other possible interpretations of soundness. This clarification is reinforced in later sections of the manuscript, where we consistently use phrases such as “soundness under temporal matching”/“From a temporal soundness perspective” to maintain specificity and transparency. We believe these changes address the concern and help readers better understand the scope and limits of our argument.
Comment 3: Avoid the ambiguous use of the term “excess” in reference to green electricity production	
Some more minor points: - The authors in a few spots refer to green claims based on “excess” production. Excess to what? If the power is being claimed in RECs that means the power was used, it wasn’t excess to the system,	Thank you for this helpful clarification. We have carefully reviewed the manuscript and removed all instances where we previously referred to green electricity claims based on “excess” production. We agree that the term was not sufficiently defined and fostered misunderstandings.

it was excess to the buyer's profile, which means the power is only "excess" as defined in your very particular framework.	
Comment 4: Revise loaded research question to neutrally reflect the scope of the analysis	
- The authors took a step in the right direction with the reframed focusing question "Would green electricity claims still stand under stricter temporal matching?" But the question is loaded. We have no reason to suspect that claims would "still stand" if we change accounting requirements. The question should how new principles would affect compliance, not if.	Thank you for pointing out that our previously stated research question could be interpreted as loaded. We agree with your assessment and have revised the phrasing accordingly. The updated research question now reads: "How would stricter temporal matching affect the soundness of green electricity claims? What implications might different temporal matching requirements have?" The new formulation should avoid presupposing whether existing claims would "stand" or not and instead frame the inquiry around how different temporal accounting requirements shape claim soundness.
Comment 5: Omit lines of reasoning that suggest marginal effects cannot influence supply	
- The authors wordsmithed some points about the relationship between price and impact, but maintained the message with the revised language "revenues... that only provide marginal incentives for further investments in new capacity." It is fundamentally the same point and encounters the same problems. There is nothing in economic theory to explain why additional revenues wouldn't affect supply in a highly-competitive industry like renewable energy development. Those effects are, of course, "marginal", but who decides how marginal is "enough"? In line with my comments above, I think the paper would be stronger by omitting this whole line of reasoning.	Thank you for this important point. As noted in our response to Comment 1, we have substantially shortened and revised the literature review. In doing so, we have also removed any lines of reasoning suggesting that marginal incentives are insufficient to influence supply.
Comment 6: Remove ambiguous or rhetorical references to "actual physical flows" of electricity	
- Just as an example of why the background discussion weakens the paper, consider this line: "the systematic failure to consider the actual physical flows of electricity..." Each REC represents 1 MWh of renewable electricity and its attributes as metered and	Thank you for this insightful comment. We acknowledge that the phrase "actual physical flows of electricity" can be misleading and rhetorically charged, particularly in the context of how RECs function within interconnected grid systems. In retrospect, we recognize that we were influenced by the language used in prior literature. However, as you

“physically” delivered instantaneously on the grid where it was generated. Was that electricity not “physical”? If “actual physical flow” implies the flow of electricity “from” a generator “to” the end user then the only way to “consider the actual physical flows of electricity” is to disconnect from the grid and only claim use off of an islanded system. Again, as I mentioned in my last round of comments, I realize that this sort of rhetoric is common, but as a referee I want good research like this paper to be published without these straw-man arguments.	rightly point out, such rhetoric can undermine analytical clarity and introduce straw-man arguments. As mentioned in our response to Comment 1, we have substantially shortened the literature review and revised its tone. In doing so, we have also explicitly refrained from using rhetoric that, while common in the literature, may be problematic from a neutral and analytically sound perspective. Any passages referring to “actual physical flows” have been removed, and the revised manuscript now frames the discussion in more precise and neutral terms, aligned with the accounting focus of our analysis.
Comment 7: Remove compliance-related language in relation to voluntary markets	
- There are a few spots where the authors discuss the voluntary market in compliance terms, most notably in the conclusion where the authors ask “whether and which penalties should occur in case of non-compliance.” No such penalties exist, this is a voluntary market, and that is fundamentally the problem. We can ask corporations to report in certain ways, maybe we can even require certain accounting practices, but we cannot require corporations to change their operations according to those practices.	Thank you for pointing this out. We agree that discussing “compliance” and “penalties” in the context of a voluntary market was inappropriate and potentially misleading. In the revised manuscript, we have removed all compliance-related language in relation to the voluntary market, including the reference in the conclusion to potential penalties for “non-compliance.” As noted also in our response to Comment 1, we have placed greater emphasis on the voluntary nature of the market throughout the revised text. Rather than implying enforceable obligations, we now explicitly acknowledge the open-ended nature of voluntary participation and the challenges this poses for ensuring adherence to more granular matching practices. In terms of future research, we thus no longer refer to penalties, but instead point to the need for exploring mechanisms that could help incentivize adherence to more stringent matching criteria under voluntary conditions. We believe that this reorientation is more consistent with the governance reality of the voluntary market.
To summarize, I like the insights this paper provides and I would like to see it published, but I can’t endorse this paper as is given the significant overreach of the assertions beyond the scientific evidence presented in the paper.	Thank you for acknowledging the contribution of our paper and for guiding us toward a clearer and more focused presentation. We have revised the manuscript to better align with the evidence presented, removing overstatements about impact and ensuring a consistent, accounting-based framing. We are very grateful for your thoughtful and constructive feedback. It has significantly improved the manuscript, and we hope the revised version now meets your expectations.

Letter to Reviewer 3: Point-by-Point Response by the Authors as of July 30, 2025

The authors have addressed all of my comments satisfactorily. I believe the paper has improved significantly, particularly in clarifying the relevance and implications of the lack of temporal matching. I could not access the link to the code.	Dear Reviewer, Thank you once again for your thoughtful and constructive feedback throughout the review process. We truly appreciate your positive assessment of the latest revision and your recognition of the improvements made. Alongside your last comment, we received additional requests from another reviewer. We, however, ensured that all of your previous suggestions remained fully addressed, as they have played a key role in improving the clarity and quality of the paper. For better accessibility of our code, we have further set up a GitHub repository: https://github.com/HannaFScholta/SheddingLightOnGreenElectricityClaims We are grateful for your comments and the positive impact your input has had on our manuscript. With best regards, The authors
--	--

Letter to Reviewer: Point-by-Point Response by the Authors

I thank the authors for taking my comments seriously and taking the time to address my points in their revisions. As the reviewers can probably tell, this is an important topic to me, and an area where I feel that mischaracterizations have been perpetuated in the literature. I remain skeptical of the implied connection between "soundness" and impact, but the authors have sufficiently downplayed the connection to the point that it no longer poses a fundamental issue for the paper, in my view. The study offers interesting insights and a useful summary of the debate, in general.	We would like to thank you for your careful and constructive review of our manuscript, and we truly appreciate the time and effort you have dedicated to providing such detailed feedback. We are also grateful for your recognition of the revisions we have implemented and the value of our study. Below, we provide a focused response outlining how we have addressed your remaining suggestion in the revised manuscript. In addition, we have implemented several minor formatting changes in response to editorial requests. These do not affect the content of the paper, but for the sake of transparency, we also provide an overview below, following the content-related revision.
Suggestion to address ambiguity of “soundness”, to situate our chosen perspective within the broader range of possible interpretations, and to further clarify its distinction from impact-oriented assessments.	
If there is time to address one suggestion, I would still recommend to define "soundness." Again, I don't think there is anything inherently unsound about annual matching. Any accounting approach only becomes unsound if we decide we want to use it for a different purpose. The annual rainfall of a country, for instance, may be a sound way of describe how rainy a country is in aggregate, but an unsound way of predicting how likely it is to rain on a specific day. Soundness, in this study, implies a view of accuracy based on physical electricity flows. It's fine to explore that view, but it is not universal, and available research does not support the premise that that specific view of soundness will enhance the impact of emissions accounting exercises.	Thanks for pointing out that the term “soundness” was still not sufficiently clear. We agree that our approach should not be mistaken for a universal definition, nor conflated with an impact-oriented perspective on emissions accounting. To address this, we have extended the introduction to explicitly acknowledge the ambiguity inherent in green claim assessments and to clarify that our study adopts temporal matching as one possible accounting principle among others. Beyond these clarifications, we have also revised our wording throughout the manuscript: Based on your comment, we have concluded that the term “soundness”, which we used as a shorthand term for temporal transparency and reliability from an accounting perspective, is too vague and could invite subjective interpretation and judgment also in an implicational direction. We have, therefore, avoided its usage and instead employ more direct language (e.g.: “more temporally aligned claims” instead of “temporally sound claims”; or “from a temporal transparency perspective” instead of “from the perspective of claim soundness”). Instead of referring to “implications for the soundness of green electricity claims,” we tried to be more precise and clarify that, from an accounting perspective, we evaluate the implications that temporal matching has on green electricity claims themselves. In line with these revisions, we clarify our perspective in the introduction. Similar to your rainfall analogy, we emphasize that the appropriateness of any accounting lens depends on the context and purpose:

It is important to note that the transparency and reliability behind such claims may be judged differently depending on the applied accounting principles. In this study, we adopt a perspective that links green claims to the availability of renewable electricity at the time of consumption, given that global decarbonization efforts aim to ultimately align total energy demand with renewable supply at all times. This temporal alignment of renewable demand and renewable generation is of high importance, especially due to the intermittency of renewable generation and the absence of free energy storage. However, we note that our perspective may not be mistaken for a universal definition. Possible alternative accounting principles that could be used in assessing green claims include, for example, adherence to recognized standards, long-term decarbonization outcomes, or investment effects.

We sincerely thank you for the dedication and thought you invested in your review. Your comments have been invaluable in sharpening the focus and improving the precision of our paper.

Overview of journal-required formatting changes (non-content related)

- Rephrased title to avoid pun and punctuation
- Removed section numbering and ensured section headings are within the character limit
- Removed footnotes and integrated their content into the main manuscript text or the supplementary material.
- Adapted the final paragraph of the introduction to include a brief summary of findings, beginning with "In this work."
- Restructured the manuscript to follow the prescribed Nature Communications order (Introduction, Results, Discussion, Methods)
- Clustered figures into panels where required and expanded figure legends to ensure figures can be understood without reference to the main text
- Updated bibliography format to fully comply with journal guidelines